



# Can Weather Patterns Contribute to Predicting Winter Flood Magnitudes Using Machine Learning?

Emma Ford[1,2], Manuela I. Brunner[3,4,5], Hannah Christensen[1], and Louise Slater[2]

[1]Atmospheric, Oceanic and Planetary Physics, University of Oxford, UK
[2]School of Geography and the Environment, University of Oxford, UK
[3]Institute for Atmospheric and Climate Science, ETH Zurich, Zurich, Switzerland
[4]WSL Institute for Snow and Avalanche Research SLF, Davos Dorf, Switzerland
[5]Climate Change, Extremes and Natural Hazards in Alpine Regions Research Center CERC, Davos Dorf, Switzerland

**Correspondence:** Emma Ford (emma.ford@hertford.ox.ac.uk)

**Abstract.** Fluvial floods pose severe socioeconomic and environmental risks globally, and are projected to change in frequency and severity in future decades. While it is crucial to understand these changes, the prediction of extreme events remains a significant challenge. Identifying predictable features driving extreme flood events provides a potential way forward with respect to improving such predictions. Weather patterns tend to be more stable and predictable than meteorological catchment-scale variables such as precipitation. However, the contribution of weather patterns to extreme flood prediction remains poorly understood. This study investigates the role of weather patterns, along with other sets of predictors, in influencing winter flood magnitudes above the 99th percentile within a large-sample machine learning framework, using natural benchmark catchments from the UK National River Flow Archive. Six generations of random forest models, each generation including additional sets of features, are explored on the national, regional, and catchment scale. Model results are interpreted using Shapley Additive Explanations (SHAP) to understand feature importance. Additionally, we analyze the conditional probabilities of the UK Met Office's MO-30 weather patterns during extreme flood events. Our findings show that weather patterns with cyclonic low pressure systems frequently co-occur with high flow magnitudes, which is also reflected in the SHAP value analysis. However, the predictive power of these weather patterns is limited and offers hardly any benefit. We also show regional nuances in the feature importance of predictors and model performance. The majority of the predictability comes from meteorological variables and antecedent precipitation. Our findings highlight the variability in model outcomes depending on the model structure and choice of predictors. This study also offers methodological guidance for developing large-sample machine learning models for flood estimation that integrate atmospheric predictors with traditional hydro-meteorological and geographical variables.





## 1   Introduction

Fluvial floods are generated by complex interactions between atmospheric, hydrological, and land-surface processes occurring across various temporal and spatial scales (Berghuijs et al., 2019; Tarasova et al., 2023; Bertola et al., 2020; Nied et al., 2014). The predictability of these events varies significantly due to the interplay of local and large-scale drivers, and the rarity of extreme events in observational records further complicates predictive modeling (Blöschl et al., 2019; **?**; Brunner and Slater, 2022). At the catchment scale, extreme flood event data are even more limited, increasing uncertainty in both physical process understanding of flood generation mechanisms and the development of robust prediction frameworks (Yuan and Lozano-Durán, 2024; Sillmann et al., 2017; Tabari, 2021).

Flood event generation is influenced by a combination of predictors operating at different timescales. The complex mechanisms responsible for flood generation are often simplified and categorized into intense rainfall, saturated soil conditions, snowmelt, and rain on snow (Liu et al., 2022; Blöschl et al., 2019; Berghuijs et al., 2019). In addition, identifying the exact time-lag of different predictors and their relative importance for driving floods is complex (Vivoni et al., 2006). This predictor time-lag is often described by capturing antecedent conditions, such as antecedent soil moisture or precipitation, over the days and months prior to a flood peak (Staudinger et al., 2024; Bennett et al., 2018). While the main drivers of flood occurrence have been well studied, disentangling the individual and combined influences of predictors across different timescales is challenging (Scussolini et al., 2024; Massari et al., 2023; Bárdossy and Filiz, 2005).

Advancing our understanding of driver importance and improving our ability to model and predict extreme floods is critical. However, capturing the dynamics and quantifying the driving features of extremes remains challenging due to the complexity of flood systems and the limited availability of extreme event data. Emerging tools such as machine learning (ML) and explainable AI (XAI) offer significant potential for exploring driver contributions, as they have been successfully applied in hydrological studies to analyze large datasets and uncover insights into hydrological systems (Ley et al., 2024; Slater et al., 2024b; Jiang et al., 2022; Coxon et al., 2024; Hamel and Brunner, 2024). Despite their promise, large-sample hydrological ML models have generally not incorporated synoptic-scale weather patterns as predictive features alongside land-surface and hydrometeorological variables, even though these features are closely interlinked (Brunner and Dougherty, 2022; Schlef et al., 2019; Prudhomme and Genevier, 2011; Duckstein et al., 1993). Furthermore, previous studies have not adopted a modeling approach that incrementally adds feature sets across generations to evaluate their contribution. This approach could be particularly helpful in quantifying the predictive skill of different feature sets for extreme flood magnitude prediction, thereby providing insights into the possible physical mechanisms and interactions that drive flooding.

Atmospheric circulation patterns are valuable tools for exploring the relationship between an event and atmospheric conditions over a large area, such as the UK and Europe (Lavers et al., 2012, 2020; Schlef et al., 2019; Bárdossy and Filiz, 2005; Duckstein et al., 1993; Wilby, 1993; Brunner and Dougherty, 2022). Weather patterns (WPs) are static categories of these atmospheric conditions defined over specific spatial and temporal scales, typically derived from meteorological variables such as mean sea level pressure (Neal et al., 2016; Lamb, 1972; Beck and Philipp, 2010). They can be distinguished from weather regimes (or types) based on their spatio-temporal scale. For example, a cyclonic low-pressure system influencing a regional





area is a weather pattern, whereas the North Atlantic Oscillation (NAO), which operates over a much larger spatial-temporal
scale, represents a weather regime (Fabiano et al., 2021; Neal et al., 2016). WPs provide a framework for exploring past,
present, and future changes in flood events under different climate scenarios (Pope et al., 2021; Khanam et al., 2024; Brunner
and Dougherty, 2022).

Since 2016, there have been advancements in associating daily WPs with extreme event phenomena. The Met Office's
MO-30 WPs produced by Neal et al. (2016) categorise daily synoptic scale UK/European weather into one of thirty discrete
weather types. They have been used to enhance our understanding of synoptic-scale atmospheric influences on coastal flooding,
precipitation extremes, drought, and temperature related mortality, and future WP frequency changes under climate scenarios
(Richardson et al., 2018, 2020; Neal et al., 2018a; Pope et al., 2021; Huang et al., 2020; Neal et al., 2024). The Met Office tool
'Decider,' presented in Richardson et al. (2018), explores flood risk by analyzing the relationship between WPs and precipi-
tation. While the tool provides valuable insights, it has not been developed to directly link WPs with catchment streamflow.
This presents a gap in understanding the MO-30 WPs' potential as predictors of fluvial flood characteristics. Furthermore, no
studies have investigated the use of MO-30 WPs as predictors of UK flood magnitudes within a large-sample machine learning
framework. Although previous research has investigated the relationship between atmospheric circulation and river flow across
Europe and the United States (Brunner and Dougherty, 2022; Schlef et al., 2019; Duckstein et al., 1993; Wilby, 1993), the
potential of weather patterns for flood prediction remains understudied.

Recent years have seen rapid developments in ML techniques that can handle non-linear interactions, large datasets, and high
variability in predictors (Fleming et al., 2021; Nevo et al., 2022; Slater et al., 2024b). These methods are particularly valuable
for analyzing complex environmental processes (Huntingford et al., 2019; Eyring et al., 2024). For example, prior work has
employed Random Forests (RF) and XAI tools to highlight feature importance in hydrological studies (Slater et al., 2024a;
Jiang et al., 2022; Xu et al., 2024). Data-driven approaches can reveal the potential drivers of extreme events, quantify their
contribution to predictions, and show how these contributions vary with event extremeness (Xu et al., 2024; Mushtaq et al.,
2024; Hamel and Brunner, 2024; Coxon et al., 2024; Slater et al., 2024b).

RF models have proven particularly effective for understanding the drivers of hydrological events due to their ability to
model complex, non-linear relationships and handle multiple data types, while avoiding overfitting (Hamel and Brunner, 2024;
Slater et al., 2024b; Jiang et al., 2022; Xu et al., 2024). RF models are an ensemble method that construct multiple decision
trees, aggregating predictions across these trees to provide robust outputs and reducing vulnerability to overfitting (Breiman,
2001; Cutler et al., 2012; Fawagreh et al., 2014). They are suited for extreme event analysis due to their resilience in data-sparse
scenarios and their interpretability compared to deep learning methods, such as Artificial Neural Networks (ANNs) (Audemard
et al., 2021). While ANNs, and more recently Long Short-Term Memory (LSTM) networks, have been extensively explored
for predicting streamflow time-series, including extreme events (Lees et al., 2021, 2022; Kratzert et al., 2018, 2022; Frame
et al., 2022), these methods typically focus on time-series predictions and are best suited for data-rich scenarios. In contrast,
our approach focuses on estimating the magnitude of extreme streamflow events above the 99th percentile threshold rather
than modeling continuous time-series data. This aligns better with the capabilities of RF models, which are well-suited for
characterizing extreme flood events in the context of limited observations of extremes.





Current flood prediction research often ignores the interaction between atmospheric and hydrological variables for flood magnitude prediction. Testing the integration of synoptic-scale features in a predictive framework is important, as novel and creative ways to enhance extreme flood prediction are needed. This study fills these research gaps by investigating the contribution of synoptic-scale weather patterns, meteorological and physical catchment features as drivers of winter extreme flood magnitudes in UK natural catchments, using large-sample machine learning random forest models.

We explore three main areas: [1] The conditional probabilities of MO-30 WPs on fluvial flood days. [2] The distributions of flood magnitudes associated with MO-30 WPs. [3] The integration of MO-30 WPs into a large-sample machine learning framework to assess their influence on flood magnitudes in UK natural catchments, alongside predictors such as geographical information, catchment characteristics, and meteorological variables. To achieve these objectives, we employ carefully crafted random forest models, running six model generations for a national model (UK pooled catchment sample) and seven regional models (catchments grouped by climate regions). Each generation incrementally includes additional predictive feature sets, including WPs. While the regional models assess the value of training a large-sample model on data with similar characteristics, relative to the full dataset (i.e., UK national model). This study advances the growing body of research on large-sample flood prediction using machine learning techniques by incorporating both atmospheric and hydrological drivers, and provides insights into the estimation of flood magnitudes above the 99th percentile. Additionally, we compare model feature importance with conditional probabilities and use explainable AI techniques to interpret the ML model outputs.

## 2 Data and Methods

### 2.1 Data Sources

Daily streamflow data between 1969 - 2021 were obtained for UK benchmark catchments from the National River Flow Archive (NRFA), as described in Harrigan et al. (2018). The corresponding catchment-averaged precipitation was also obtained from the NRFA, and this was originally sourced from the CEH-GEAR dataset (Keller et al., 2015). This temporal span of 52 years was chosen due to the availability of data. Benchmark catchments are considered to be as minimally influenced by anthropogenic influences as possible, and are smaller than other catchments, with a median size of 100 km$^2$ and are a suitable dataset for long-term analysis of floods (Harrigan et al., 2018). Benchmark catchments were chosen to explore the influence of the WPs as predictors without the influence of noise from anthropogenic activity as far as possible. Moreover, we benefit from the length of their time-series, compared to other available datasets. From the benchmark catchments, we selected those for analysis based on the following criteria: at least 95% of data must be available for each water year (October 1st - September 30th) $y$, and at least 30 years of data must meet the former requirement. For each unique catchment, the normalization of streamflow (m$^3$/s) to specific discharge (mm/day), accounted for catchment size variability in the large-sample model, and enhanced model generalizability.

These selection criteria resulted in 134 suitable catchments. The CAMELS-GB dataset (Coxon et al., 2020) provided catchment averaged precipitation, potential evapotranspiration, temperature and static catchment variables for each of the selected catchments. The full list of variables extracted and any calculated lagged variables are displayed in Table 1. For exploration of



a UK sample, all selected catchment data were pooled together. For regional analysis, the catchments were grouped into their corresponding Met Office Hadley Centre Observations Dataset (HadUKP) climate regions using shapefiles provided by the Met Office (Met Office, 2023). Daily WP classifications were obtained from the Met Office MO-30 dataset (Neal et al., 2016),
for the same time period as the catchment data (1969–2021). These WPs were created using an annealed k-means clustering method of mean sea level pressure (MSLP) from the European Mean Sea-Level Pressure dataset (EMSLP) (1850–2003) (Neal et al., 2016; Ansell et al., 2006). Lower-numbered WPs are associated with weaker MSLP anomalies, are historically more frequent, and occur more often in summer. Higher-numbered WPs are associated with stronger MSLP anomalies, are historically less frequent, and occur more often in winter (Neal et al., 2016). The WPs are displayed in Figure 1, and their corresponding
descriptions from the Neal et al. (2016) dataset are presented in Supplementary Information Table A1



**Table 1.** Variables sourced for the machine learning models, covering the period 1969 - 2021.

| Variable | Units | Source | Description |
|---|---|---|---|
| Specific Discharge | mm/day | NRFA, 2023 | Measured as streamflow ($m^3$/s) at gauging stations; converted to specific discharge using catchment area. |
| Latitude | Degrees | NRFA, 2023 | Latitude of the catchment. |
| Longitude | Degrees | Harrigan et al. (2018) | Longitude of the catchment. |
| Daily Precipitation | mm/day | Keller et al. (2015) | Catchment-averaged daily precipitation originally sourced from the CEH-GEAR dataset. |
| Lagged Precipitation | mm/day | Keller et al. (2015) | Precipitation for the day before, two days before, and three days before (separately) to capture antecedent moisture conditions. |
| Area | $km^2$ | Coxon et al. (2020) | Total catchment area. |
| Aridity Index | - | Coxon et al. (2020) | Ratio of mean daily potential evapotranspiration to mean daily catchment-averaged precipitation. |
| Runoff Ratio | - | Coxon et al. (2020) | Ratio of mean annual streamflow to mean annual precipitation. |
| Streamflow Elasticity | - | Coxon et al. (2020) | Sensitivity of streamflow to precipitation changes. |
| Baseflow Index | - | Coxon et al. (2020) | Proportion of baseflow in total streamflow. |
| Urban Land Cover | % | Coxon et al. (2020) | Percentage of urban land-surface cover in the catchment. |
| Maximum Elevation | m | Coxon et al. (2020) | Maximum elevation for the catchment. |
| Potential Evapotranspiration | mm/day | Coxon et al. (2020) | Catchment-averaged daily potential evapotranspiration (PET). |
| Daily Temperature | °C | Coxon et al. (2020) | Daily mean, minimum, and maximum temperatures. |
| Weather Pattern Categories | - | Neal et al. (2016) | Daily classification (1–30) based on MSLP anomalies from Ansell et al. (2006). |
| Lagged Weather Patterns | - | Neal et al. (2016) | Weather pattern categories for the day of flood magnitudes, one day before, two days before, and three days before to capture antecedent conditions. |





**Figure 1.** Weather pattern classification defined by Neal et al. (2016) created from European Mean Sea-Level Pressure dataset for 1850–2003, with a 5° spatial extent as described in Ansell et al. (2006). The mean sea level pressure anomalies for each pattern are plotted at 2 hPa intervals. Reproduced from Neal et al. (2016).

## 2.2 Identifying Extreme Flood Magnitudes

The winter months December, January, February (DJF) were chosen for analysis because most of the largest flood events in the UK occur during this season (Ledingham et al., 2019). Focusing on winter also simplified the analysis by eliminating inter-seasonal variability. The flood events exceeding the 99th percentile were identified individually for each catchment between 1969 and 2021 using the statistical Peak Over Threshold (POT) method (Rosso; Rodding Kjeldsen and Prosdocimi, 2023a, b).





To ensure independence between events, a 7-day time-lag was applied, following Brunner and Dougherty (2022). The target variable for the machine learning models (flood magnitude) was defined as the largest specific-discharge value (mm/day) for each independent event. The POT method is widely recognized for its ability to capture multiple extreme events within a single year, providing a more comprehensive analysis of flood extremes compared to the Annual Maxima (AM) method. A threshold

set at the 99th percentile was used to focus on the largest events while ensuring a sufficient sample size for machine learning applications. Static attributes, such as catchment characteristics and geographical variables (e.g., latitude and longitude), were included as predictors in the models and remained constant throughout the period of record.

For meteorological variables, the values corresponding to the identified flood magnitude days were extracted. Additionally, antecedent cumulative precipitation and WP categories were included for the three days preceding each flood event to capture

critical synoptic-scale circulation and antecedent moisture conditions, both essential for understanding the factors leading to extreme floods. The choice of a 3-day antecedent period was informed by the small size and natural conditions of the catchments analyzed. For the UK analysis, the natural catchments are pooled into one large-sample. For the regional analyses, the HadUKP regional shapefiles provided by the Met Office were used to group the natural catchments into distinct climate regions based on precipitation characteristics (Richardson et al., 2018; Met Office, 2023). See Table2 for more information.

**Table 2.** Total event count breakdown by region, ordered from highest event count (SW) to lowest (NS). Event count is the number of independent flood magnitude events exceeding the 99th percentile during Boreal winter (DJF) between 1969 - 2021 for the 134 selected natural catchments in each region and the UK overall. The shortened names (e.g., Southeast England as 'SE') will be used throughout the paper. The Number of catchments and the average event count per catchment within each region is also noted.

| Region | Total Event Count | Number of Catchments | Catchment Average Event Count |
|---|---|---|---|
| Southwest England and South Wales (SW) | 1828 | 33 | 14 |
| Central and Eastern England (CEE) | 1018 | 20 | 13 |
| Northwest England and North Wales (NW) | 1061 | 19 | 16 |
| Southeast England (SE) | 730 | 17 | 12 |
| Northeast England (NE) | 694 | 14 | 13 |
| South Scotland (SS) | 687 | 12 | 18 |
| East Scotland (ES) | 511 | 11 | 16 |
| North Scotland (NS) | 405 | 8 | 13 |
| **UK Overall** | **6934** | **134** | **14** |



## 2.3 Conditional Probabilities

To identify dominant weather patterns for flood development, we explored the WPs most frequently associated with extreme flood events and computed the probability of each WP given that a flood event occurred. This analysis was conducted for (1) all catchments combined and (2) catchments grouped by HadUKP climate regions (Met Office, 2023). Since the dataset is pre-filtered to contain only extreme flood magnitude days, this is an examination of the conditional probabilities, meaning the frequency of WPs given a flood magnitude has occurred. To account for the cumulative influence of synoptic-scale conditions leading to floods, the method extended the analysis beyond the day of the event to include lagged WPs (up to three days prior).

### 2.3.1 Distribution of Flood Magnitudes

We further explored the distribution of flood magnitudes associated with each WP, and present this for the WPs most often associated with flood magnitude days. Flood magnitude (mm/day) has already been normalised. However, since larger catchments might represent smaller flood magnitudes compared to smaller catchments, given their larger drainage area, we present the flood magnitudes associated with each WP stratified by catchment size. The catchment sizes were categorized into 'Small' ($3.12$–$66.82$ km$^2$), 'Medium' ($66.82$–$194.81$ km$^2$), and 'Large' ($194.81$–$1505.54$ km$^2$) using three bins containing approximately one-third of the data each.

## 2.4 Machine Learning Model Structure

We developed six different generations of RF regression models to gain insights about the roles of the different feature sets as predictors of extreme flood magnitudes (mm/day), for the UK and regional samples (see Table 3). The change between generations is the incremental addition of different feature sets. We choose a RF model due to its ability to manage non-linear and complex relationships between the predictors and the target variable. We implement the RF using the Scikit-learn Python package by Pedregosa et al., and calculate the performance metrics and Shapley Additive Explanations on the test set results.



**Table 3.** Generations of machine learning models used with UK-wide and regional data.

| Gen. | Features Added | Physical Processes Captured |
|---|---|---|
| 1 | Latitude, Longitude | Spatial variability of flood magnitudes within and between catchments, due to geographical, topographical, and catchment characteristic differences. |
| 2 | Weather Pattern category on day of event | Synoptic-scale meteorological conditions. |
| 3 | Weather Pattern category on event day, day before, two days before, and three days before | Synoptic-scale antecedent meteorological conditions at different time-lags. |
| 4 | Static Catchment characteristics | Influence of catchment properties on flood dynamics. |
| 5 | Catchment averaged precipitation, temperature (mean, maximum, minimum), and potential evapotranspiration on the day of event | Role of local meteorological variables. |
| 6 | Accumulated catchment averaged precipitation on the day before, two days before, and three days before event | Antecedent local moisture conditions to capture the influence of different time-lags of catchment wetness. |

One-hot encoding (1 = TRUE, 0 = FALSE) was applied to the WP categories to create binary features for the model. Model hyper-parameters were optimized using RandomizedSearchCV. A sensitivity analysis indicated that increasing the complexity of the model parameters (e.g., 1000 trees compared to 2000 trees, and changing the minimum number of splits) did not substantially impact model performance. The model hyper-parameters chosen from RandomizedSearchCV were `n_estimator = 1000`, `min_samples split = 10`, `min_samples leaf = 2`, `max_depth split = None` and `Bootstrap`

`= TRUE`. The RF model choice allowed us to leverage the benefits of ensemble regression trees to improve predictive accuracy and increase robustness of models (Slater et al., 2024b; Hamel and Brunner, 2024; Coxon et al., 2024). The `Bootstrap = TRUE` introduces randomness which helps reduce overfitting and limiting correlation between trees (Breiman, 2001, 1996). The same hyper-parameters were used across all model generations to allow for the comparison of changing the feature sets.

A temporal train-test split was used for the UK and regional models. Specifically, 80% of data (1969 - 2010) was used for

training and 20% (2011 - 2021) for testing. This temporal validation method has been used in other recent hydrological studies such as Jiang et al. (2022), and respects the temporal sequence of the data, preventing leakage between training and testing sets. This allows for an assessment of model performance on unseen data, as would be the case in a realistic forecasting scenario (Botache et al., 2023).By applying SHAP to the test set, we can understand how the model behaves on unseen data, providing insights into its generalization capabilities and feature importance in the testing period. Additionally, this analysis provides

valuable insights into the influence of different feature sets on model predictions, particularly in the most recent period of the





time series. Whilst this temporal validation does not directly address spatial generalization, this is mitigated to an extent by the grouping of catchments into regional models, allowing for spatial insights. The described model architecture in this section remained consistent for the UK and regional models, with the only difference being the incremental addition of features to represent new physical processes per model generation.

## 2.5 Test Set Evaluation Metrics and SHAP Calculation

For each model generation, test set predictions were evaluated using performance metrics $R^2$ and Percentage Bias (PBIAS). These metrics were calculated for the UK and regional results, and $R^2$ was also calculated at the individual catchment level. Catchment-level calculations were limited to catchments with more than 10 events in their test set to ensure reliable performance assessment. This approach enabled evaluation of model performance across spatial scales.

The equations for the performance metrics calculated on each test set per model generation are as follows:

$R^2$ (**R-squared)** measures the proportion of variance in actual flood magnitudes captured by the model predictions. $R^2$ values typically range from 0 to 1, with values closer to 1 indicating a better fit. $R^2$ was assessed at the national, regional, and individual catchment levels. The formula is as follows (Chicco et al., 2021):

$$R^2 = 1 - \frac{\sum_{i=1}^{n}(y_i - \hat{y}_i)^2}{\sum_{i=1}^{n}(y_i - \bar{y})^2}$$

where $y_i$ is the observed value, $\hat{y}_i$ is the predicted value, and $\bar{y}$ is the mean of observed values.

**Percentage Bias (PBIAS)** evaluates whether the model tends to overestimate or underestimate the observed values. In this definition, a negative PBIAS indicates underestimation (predictions are lower than observed values), and a positive PBIAS indicates overestimation (predictions are higher than observed values). The formula, consistent with the revised convention, is as follows (Towler et al., 2023):

$$\text{PBIAS} = \frac{\sum_{i=1}^{n}(\hat{y}_i - y_i)}{\sum_{i=1}^{n} y_i} \times 100$$

where $y_i$ is the observed value and $\hat{y}_i$ is the predicted value.

### 2.5.1 Significance of $R^2$ change across model generations

To evaluate the incremental effect of feature set additions on model performance across generations, paired permutation randomized significance tests were conducted for the $R^2$ results. This method assesses whether changes in $R^2$ are statistically significant between successive model generations, or when compared to the baseline model (Generation 1 using latitude and longitude). Flood event predictability inherently varies due to hydro-meteorological conditions and the physical characteristics of catchments (Brunner et al., 2021; Hakim et al., 2024), which can introduce day-to-day and catchment-to-catchment variability in model performance. Permutation testing, a non-parametric approach widely used in machine learning and environmental sciences, provides a robust framework for evaluating the statistical significance of observed effects without assuming data normality (Graham et al., 2014; Ojala and Garriga, 2010). It is particularly useful in cases with small sample sizes or heterogeneous data (Nariya et al., 2023; Graham et al., 2014).





In this study, paired permutation testing was used to assess whether changes in model $R^2$ across generations were the result of feature set changes or random variability. First, for a given region, predictions for the same flood events were extracted across model generations, and the observed difference in $R^2$ between two model generations was calculated as:

$$\Delta R^2_{\text{obs}} = R^2_{\text{Gen}_{j+1}} - R^2_{\text{Gen}_j}.$$

To simulate the null hypothesis ($H_0$), which assumes no systematic difference in $R^2$, model predictions for the same flood events were randomly shuffled between the two generations. The shuffled predictions were used to recalculate $R^2$ for each generation, and the difference was computed as:

$$\Delta R^2_{\text{shuffled}} = R^2_{\text{shuffled, Gen}_{j+1}} - R^2_{\text{shuffled, Gen}_j}.$$

This shuffling process was repeated 1000 times to construct a null distribution of $\Delta R^2$ differences. Finally, the $p$-value was calculated as the proportion of shuffled differences that were as extreme as or more extreme than the observed difference:

$$p = \frac{\sum_{b=1}^{B} |\Delta R^2_{\text{shuffled},b}| \geq |\Delta R^2_{\text{obs}}|}{B}.$$

If $p < 0.05$, the observed change in $R^2$ was considered statistically significant, indicating that the feature set changes had a meaningful effect on model performance.

This analysis was applied to each region and the UK samples for model generations 1 to 6. By pairing predictions for the same flood events, the method controlled for variability in event predictability, allowing robust comparisons across generations. Importantly, this approach does not assume normality, making it suitable for datasets with limited or heterogeneous samples. The significance testing provided rigorous insights into $R^2$ improvements across regions and model generations.

### 2.6   Model Interpretability with SHAP (SHapley Additive exPlanations)

SHAP, derived from cooperative game theory, is a powerful explainable AI tool used to interpret the outputs of machine learning models (Lundberg and Lee, 2017). SHAP values quantify each feature's contribution to individual predictions, providing insights into local and global influences on the target variable (Lamane et al., 2024). We calculated SHAP values for each regional and generational model to understand the role of individual features in predicting flood magnitude. This approach enhances our understanding of the RF decision-making process, and the relative importance of features. For each prediction in

the test set, SHAP values represent the extent to which a feature contributes to deviations from the mean prediction (Wang et al., 2016; Xu et al., 2024). Compared to feature importance derived from Gini impurity, SHAP is more robust as it provides insights into both the direction and magnitude of the relationship between predictors and the target variable, and enables analysis of local and global effects (Lundberg et al., 2020; Lundberg and Lee, 2017). As presented in Lundberg et al. (2020), Lundberg and Lee (2017), and (Xu et al., 2024), SHAP can be explained by the following:

The SHAP value $\phi_i(f,x)$ for a feature $x_i$ calculates a feature's contribution to the model's prediction:

$$\phi_i(f,x) = \sum_{S \subseteq N \setminus \{i\}} \frac{|S|!(|N| - |S| - 1)!}{|N|!} \left( f(x_{S \cup \{i\}}) - f(x_S) \right)$$



where $\phi_i(f, x)$ represents the SHAP value of feature $x_i$, $f$ denotes the model's predictive function, $N$ is the set of all features, and $S$ is any subset of $N$ excluding feature $x_i$. Here, $x_S$ represents the input under the given feature set $S$, and $|N|$ and $|S|$ correspond to the sample sizes of sets $N$ and $S$.

## 3 Results and Discussion

### 3.1 Conditional Probability and Distribution Results

Figure 2 (a) shows the conditional probability of each WP given the occurrence of a flood magnitude event. WP 30, described as a cyclonic southwesterly low-pressure system by Neal et al. (2016), emerges as the most frequently associated WP with flood magnitude days across the UK (Figure 2) . In the SW, WP 30 occurs on 19% of flood magnitude days, suggesting an association. This likely reflects the physical characteristics of cyclonic systems, such as strong uplift of moist air, which can lead to intense and widespread precipitation (Cuckow et al., 2022; Richardson et al., 2018). While these findings align with previous studies also linking WP 30 to precipitation extremes and coastal flooding (Richardson et al., 2018; Neal et al., 2018b), it is critical to recognize that association alone does not capture causation or the underlying mechanisms driving flood magnitudes.

Other cyclonic WPs types, such as WP 20, WP 21, and WP 29, also show associations with flood magnitude days across the regions (Figure 2 (a)) , with certain regions such as the NS experiencing a stronger association with WP 23 than other locations. The regional nuances displayed may highlight the interplay between atmospheric drivers and localized catchment characteristics in certain UK regions. While WP 30 dominates in the SW, other cyclonic WPs exhibit higher relevance in regions where specific hydrological and topographical factors, such as soil saturation, land use, and catchment geomorphology, may be modulating flood magnitude responses Griffin et al. (2025, 2022); Berghuijs et al. (2019). These differences in observed associations between WPs and flood event magnitudes may also be influenced by the spatial variability in precipitation patterns associated with each WP. This underscores the need for further exploration of how meteorological properties of WPs interact with catchment characteristics and antecedent catchment conditions, to drive extreme flood magnitudes. Regional variability in WP-flood relationships could be further influenced by the spatial distribution of precipitation associated with each WP. For example, WP 30 is dominant in the SW and SE, where coastal and orographic influences may amplify its impact through concentrated, high-intensity precipitation. These spatial and regional variations underscore the importance of incorporating both WP-specific precipitation distributions and catchment-scale dynamics into predictive models tailored to local hydrological and meteorological characteristics. The relationship between the WPs and precipitation was previously explored by Richardson et al. (2018) and Richardson et al. (2020). The findings in Richardson et al. (2020) suggested that WP21 is associated with extreme precipitation across multiple UK regions, indicating that all regions were wetter than normal under this WP type. Similarly, the results in this study find that WP21 is also influential across many regions, with a stronger frequency in the SE, SS, and NW region on extreme flood magnitude days (Figure 2 (a)) . However, WP 30 overall stands out more on flood magnitude days, as being the most dominant type. This highlights that the WP associated with the most extreme precipitation, does not necessarily translate to the WP associated with extreme flood magnitude days across UK regions.




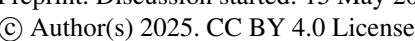

**Figure 2.** Conditional probabilities for weather patterns.(a) Conditional probabilities on flood magnitude days and (b) three days prior to flood magnitude days. Regions are indicated on the y-axis, weather patterns on the x-axis, and the color bar represents the conditional probability value between 0 and 1. Regional names shortened as in Table 2. For example, Central and East England is CEE.



Lagged weather patterns, mainly higher-numbered cyclonic types, persist in the days preceding extreme flood magnitude events. For example, three days prior to the magnitude day (see Figure 2[b]). This reflects the importance of atmospheric precursors to flooding. The dominance of WP 30 diminishes three days prior to the flood magnitude day overall, with WP 20 and 29 becoming more dominant. This suggests that multi-day atmospheric sequences, involving antecedent weather patterns, may play a key role in flood generation. These sequences likely combine with previous cumulative precipitation and soil
moisture conditions to establish conditions for flooding (Berghuijs et al., 2016, 2019).

While the influence of lagged patterns is evident, their interaction with other hydrological drivers, such as antecedent soil saturation, precipitation intensity, and hydrological memory, warrants further investigation. Existing studies (e.g., Brunner and Dougherty, 2022) highlight the critical role of antecedent conditions in amplifying flood magnitudes, emphasizing that extreme flooding often results from the interplay of atmospheric patterns and pre-existing hydrological states, rather than single weather
events. Moreover, further investigation into the persistence and sequences of WPs preceding extreme flood event magnitudes should be explored in future studies. A more holistic approach is needed to assess how sequences of atmospheric drivers interact with catchment characteristics to produce flood extremes.

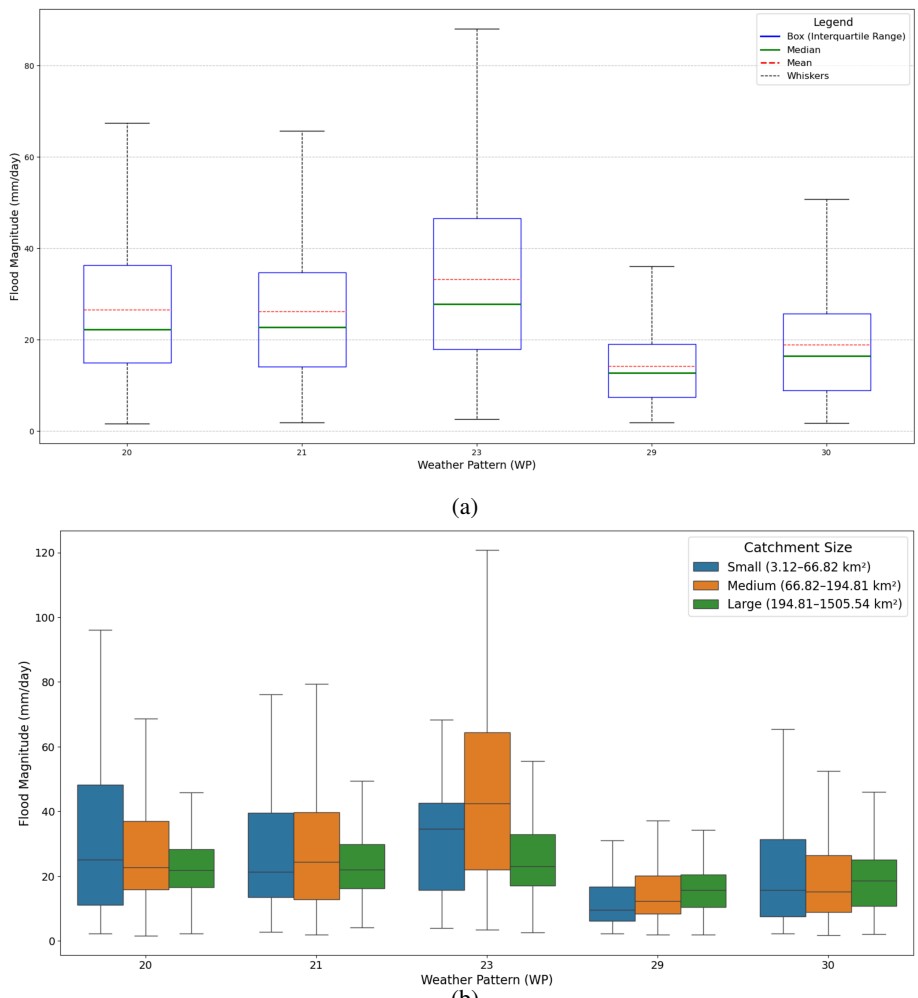

**Figure 3.** Comparison of flood magnitude (mm/day) distributions: (a) Distribution for WPs 20, 21, 23, 29, and 30 across all natural catchments. The boxplots display the IQR, represented by the blue boxes, which spans from the Q1 to Q3. The green line within each box indicates the median value, while the dashed red line represents the mean. Whiskers (black dashed lines) extend to the most extreme data points within 1.5 times the IQR from Q1 and Q3. Outliers, defined as data points beyond this range, are excluded from the plot. (b) Distribution stratified by catchment size bins for the same WPs. Catchment size bins are categorized as Small (3.12–66.82 km$^2$), Medium (66.82–194.81 km$^2$), and Large (194.81–1505.54 km$^2$).




Despite WP 30's frequent association with flood magnitude days, it is not linked to the highest mean or median flood magnitudes (Figure 3[a]). This reveals a difference between the frequency of occurrence and conditions conducive to magnitude severity. WP 23, characterized by strong winds, is associated with higher flood magnitudes than WP 30, despite its lower frequency of co-occurrence. WP 23 has a stronger association in the NS region, where catchments are observed to experience higher magnitude flood events. This suggests that WP 23 may act as a more potent trigger for severe flood magnitudes in regions like NS, perhaps when combined with antecedent conditions such as soil saturation, snowmelt, or localized topographical factors. The disproportionate impact of less frequent patterns like WP 23 highlights the need to balance frequency based WP predictions with intensity focused assessments. With projections of a 40% increase in WP 23 occurrences under the RCP8.5 climate scenario (Pope et al., 2021), its implications for flood risk in regions like NS pose significant questions about future flood risk, the need for targeted adaptation strategies, and exploring the relationship between WPs and catchment characteristics for flood generation.

Catchment size has some influence in modulating the relationship between WPs and flood magnitudes (Figure 3[b]). Smaller catchments exhibit a wider range of flood magnitudes under WPs 23 and 30, which may be explained by a sensitivity to high-intensity rainfall and shorter response times. In contrast, larger catchments tend to display a narrower range of flood responses to these patterns, as their capacity to absorb and distribute precipitation over a broader area may dissipate immediate impacts. This suggests the importance of the antecedent cumulative effect of precipitation in larger catchments, which may be more affected by prolonged or sequential rainfall events, represented by the WPs.

### 3.1.1 UK and Regional Scale ML model Performance

This section focuses on the UK and regional model results in Figure 4 (a) and (b). To evaluate the predictive performance of the RF models, $R^2$ values were calculated at the catchment, regional, and UK scales across the six model generations on the test data sets. The UK model consistently outperformed the regional models across all generations, achieving the highest $R^2$ value of 0.84 in Generation 6 (Lat, Lon + WP + CC + HM + AHM) (see Figure 4 (a)). This statistically significant improvement from the baseline $R^2$ of 0.66 reflects the advantages of pooling data across all natural catchments. The UK model's ability to capture inter-catchment variability and broader hydrological patterns highlights the benefits of using a larger, more diverse dataset for model training (Kratzert et al., 2024a; Slater et al., 2024b; Kratzert et al., 2019). The UK model's improvement in $R^2$ in later generations suggests that dynamic hydrometeorological features, such as antecedent precipitation and soil saturation, capture critical extreme flood magnitude generating mechanisms. These features likely reflect the physical processes underlying extreme flood event generation generally across the UK natural catchments, including the accumulation of antecedent rainfall leading to saturated soil conditions in the days prior to the event magnitude, which reduces infiltration capacity and increases surface runoff. This aligns with theoretical understanding that antecedent hydrological conditions are key drivers of flood magnitudes, especially in areas with variable rainfall patterns and high soil permeability (Blöschl et al., 2019; Berghuijs et al., 2019; **?**).

However, while the UK model benefits from large-sample pooled data, it may mask regional disparities in specific catchments in predictive performance. The improvement in $R^2$ in the final two generations does not necessarily indicate uniform





(a)

(b)

**Figure 4.** Comparison of UK and regional model results. The UK model is assessed on all UK catchments, while each regional model is assessed on that region's catchments: (a) $R^2$ values for different model generations (x-axis) and regions (y-axis), and (b) percentage bias (PBIAS) results for the same model generations and regions. In panel (b), blue colors indicate overestimation (positive PBIAS, predictions are too high), and red colors indicate underestimation (negative PBIAS, predictions are too low). Both figures highlight the performance and accuracy of the models as features are added incrementally (along the x-axis). In panel (a), bold and (*) indicate a statistically significant change (whether improvement or degradation) compared to the baseline Generation 1 model.





improvements at the catchment or regional level, where physical mechanisms driving flood generation may vary significantly. The $R^2$ for the UK reflects the overall aggregated performance, and does not indicate individual catchment level performance.

Although WP conditional probabilities provided association insights, their use as standalone predictors remained uncertain.
Results in Figure 4 (a) indicate that in many regions (UK, SS, NW, NE, SE) there was a statistically significant decrease in model performance with the addition of WPs as features on the day of the flood magnitude, and or antecedent days (up to three days prior) compared to the baseline model. These results suggest that not only did these features not capture extreme event flood dynamics in these regions, but they also decreased the model skill. This decline could be attributed to a combination of feature redundancy, increased dimensionality, and reduced signal-to-noise ratio, particularly in smaller regional datasets. This
suggests that the WPs may introduce redundancy or overlap with other features, potentially causing overfitting or reducing the model's ability to generalize. Furthermore, the temporal resolution of WP data may not align well with the dynamics of flood generation, further limiting their contribution. There may be an association between the WPs and extreme flood magnitude days, but the WPs do not provide enough information to contribute to the prediction of magnitude values.

Unlike the UK-wide pooled model, where feature diversity helps capture a broad range of hydrological conditions, regional
models have fewer training samples, making them more sensitive to noise and overfitting when additional predictors are introduced. A key issue is the balance between exploring feature relative importance, predictability and variability at different spatial scales. The performance of regional models varied substantially (see Figure 4 (a), reflecting differences in hydrological regimes, the number of extreme events extracted from the time-series (see Table 2) , and the explanatory power of the feature sets across region specific flood dynamics. The SW region achieved one of the highest $R^2$ values among the regional models,
with a final value of 0.83 in Generation 6. Statistically significant positive changes were observed in Generations 6 with the addition of the antecedent hydrological variables, compared to the baseline model. For example, the UK model by + 0.14, the NW by + 0.29, and the SW by + 0.13. This performance change likely reflects the regions larger dataset and hydrological characteristics, such as a high frequency of frontal rainfall events and complex topography that aligns well with the inclusion of the antecedent features. The strong response to antecedent hydrometeorological inputs suggests that soil saturation and cumulative
precipitation three days prior to the event magnitude are dominant flood generating mechanisms in this region. However, the modest statistically significant improvement from Generation 1 to 6 of + 0.13 for the UK, suggests that much of the predictive power was already captured by the baseline model, and improved only with dynamic hydrometeorological inputs.

The NW region exhibited the largest statistically significant improvement over the baseline (+ 0.29), achieving an R$R^2$ of 0.75 in Generation 6. All intermediate generations also showed significant changes (e.g., Generation 3: 0.30; Generation
4: 0.37), with the most substantial gains occurring in Generations 5 (0.69) and 6 (0.75). The region's sensitivity to dynamic predictors, such as antecedent precipitation, highlights the importance of understanding flood generating processes dominated by rapid response catchments with steeper gradients. The NW's flood dynamics are often driven by a combination of orographic rainfall and rapid surface runoff, which makes the inclusion of dynamic features particularly valuable.

Overall, the SS region showed mixed results, with statistically significant negative changes observed in Generations 2 (0.40),
3 (0.39), and 4 (0.40). The final $R^2$ of 0.59 in Generation 6, while an improvement over the baseline (0.49), may reflect the region's limited event dataset. In ES, only the final $R^2$ of 0.54 in Generation 6 showed a statistically significant improvement





over the baseline (0.46). This suggests that the included feature sets provided limited additional value, potentially due to the region's relatively homogeneous catchments or smaller dataset. CEE had the lowest baseline $R^2$ (0.28), and only the final $R^2$ of 0.37 in Generation 6 was statistically significant, despite having one of the largest datasets available. The minimal

improvements across generations indicate that again the extreme event flood dynamics were not captured by the feature sets, and potentially additional predictors or higher-resolution data may be needed to better capture flood dynamics in this region.

    Finally, the SE region exhibited statistically significant decreases in Generations 3 (0.51) and 5 (0.49). The lack of significant improvements in later generations suggests redundancy in the feature set or challenges in capturing extreme events in this region, with hydrometeorological variables such as antecedent precipitation not being as important as for the UK large-sample

model, the SW, NW, or NS regions. The SE region's relatively lower sensitivity to antecedent precipitation and hydrometeorological inputs suggests that other factors, such as urbanization and engineered drainage systems, may dominate flood generation.

    In the NW region, the inclusion of both event day and antecedent WPs significantly decreased $R^2$ compared to the baseline model. This could indicate that the NW's flood dynamics are more strongly driven by other meteorological or hydrological

factors, reducing the relevance of WP based predictors. These findings highlight the context dependent role of WPs and underscore the need for more refined temporal and spatial representations of atmospheric drivers. When comparing this to results in Figure 2, it suggests that there may be an association between WPs and extreme flood magnitudes, but the relationship is either not being captured in how the WPs have been feature engineered, or there is not a strong enough signal linking theses synoptic-scale patterns to flood magnitudes with current data limitations.

The inclusion of catchment characteristics (Generation 4) generally resulted in $R^2$ values worse than or comparable to the baseline model (Generation 1). This outcome suggests that latitude and longitude, as baseline features, may already encode significant inter-catchment variability, effectively identifying catchment-specific behavior. Adding static catchment characteristics provided limited additional value, particularly in regions where natural catchments exhibit relatively homogeneous conditions (e.g., soil type and land use) (Harrigan et al., 2018). This redundancy could explain the limited improvement observed in re-

gional models when catchment characteristics were included. Furthermore, the static nature of these features makes them less informative for modeling dynamic processes like extreme flood magnitudes, which may be more influenced by dynamic meteorological and hydrological conditions. Furthermore, as the catchments are near-natural, they may have much lower variability of catchment characteristics, such as percentage urbanization.

    Overall, the inclusion of dynamic hydrometeorological features (Generations 5 and 6) significantly improved $R^2$ for the UK,

SW, and NW models, emphasizing the importance of capturing antecedent conditions in flood prediction. However, the mixed impact of WPs (Generation 3) suggests that the inclusion of atmospheric predictors may introduce redundancy or misalignment with flood dynamics. WPs typically describe synoptic-scale atmospheric conditions but may fail to capture localized convective rainfall events or sub-daily precipitation extremes, which are critical drivers of flooding in certain regions (Lavers et al., 2012). The limited contribution of static catchment characteristics (Generation 4) further highlights the need to focus on other dynamic

inputs that capture temporal variability for extreme events, and to consider feature engineering carefully, as the information captured in the baseline generation with longitude and latitude may produce redundancy for static catchment characteristics.



The PBIAS results (Figure 4(b)) reveal systematic biases in flood magnitude predictions. In the SW and SE regions, moderate underestimation occurred consistently, with PBIAS values ranging from 7.1% to 13%. This underestimation likely reflects the model's limited ability to predict the upper tail of the flood magnitude distribution, particularly for extreme events with smaller sample sizes in the training data. In contrast, the NS region exhibited consistent overestimation (negative PBIAS), which is unexpected given the general tendency for underestimation in other regions. This discrepancy may reflect unique hydrological or meteorological characteristics in NS catchments or limitations in the feature set for this region.

The NE and NW regions displayed moderate underestimation, similar to the SW and SE regions. For the CEE region, the highest PBIAS values were observed across all generations, indicating that the model struggled to accurately predict flood magnitudes in this region. This aligns with the $R^2$ results and may reflect a combination of limited event data and distinct hydrological characteristics that are underrepresented in the feature set. In contrast, the NS region showed gradual improvements in overestimation across generations, particularly with the inclusion of antecedent conditions and precipitation features. This suggests that dynamic inputs help the model account for event-specific variability, improving its overall balance in regions with challenging hydrological dynamics.

### 3.1.2 Catchment Scale Performance

In this section, we focus on the performance of individual catchments within each of the regional and UK models. Unlike the regional and UK overall model performance discussed earlier, which combines intra- and inter-catchment variability, this analysis isolates intra-catchment variability, providing insights into how well the models capture localized flood dynamics. Figure 5, presents the distribution of $R^2$ values at the individual catchment level within the UK and regional models, contrasting intra-catchment variability against the combined performance metrics shown in Figure 4.

The SW regional model achieved high $R^2$ values at the aggregated regional scale, reflecting its ability to capture general patterns across the region, supported by a relatively larger dataset. However, at the individual catchment level, $R^2$ values varied widely (range: -0.94 to 0.69), highlighting heterogeneity in performance. This variability suggests that regional models may focus too much on learning to predict the difference between catchment behaviors, rather than predicting the behavior of a given catchment over time. They may oversimplify localized catchment dynamics, inadequately representing the variability inherent in extreme flood events. Poor performance in catchments may arise from unique hydrological characteristics, data limitations, or event-specific dynamics that reduce the relevance of certain predictors. These findings highlight the importance of feature engineering tailored to localized processes for models aiming to balance generalization and capturing local catchment dynamics.

Across all regions, individual catchments generally performed better in the UK model than in their respective regional models (Figure 4). This supports findings in the literature that large-sample pooled models leverage broader datasets to improve generalizability and robustness (Kratzert et al., 2019, 2024b; Slater et al., 2024b). The UK model achieved the highest $R^2$ values for some catchments in Generation 6 (up to 0.89). However, these high scores were limited to a few catchments, indicating that while the UK model can excel in certain cases, it does not consistently outperform regional models in terms of median or mean $R^2$ values.

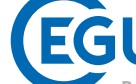



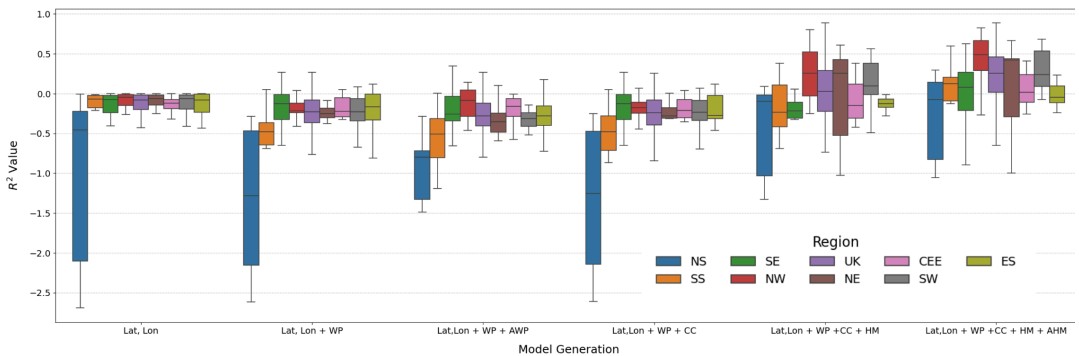

(a)

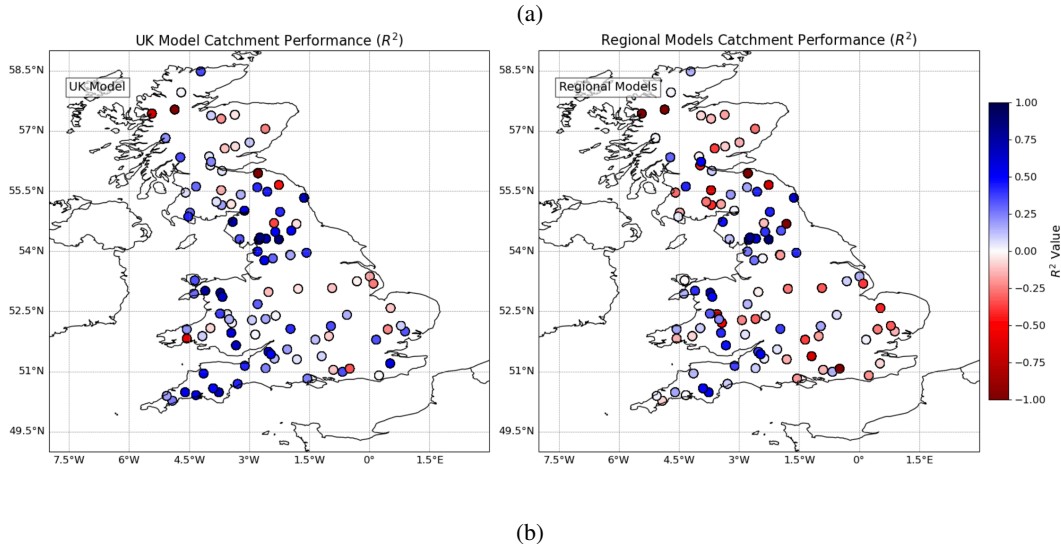

(b)

**Figure 5.** Comparison of $R^2$ catchment performance: (a) Distribution of $R^2$ for individual catchment performance within each of the regional models and the UK model. Only catchments with more than 10 test set samples are included. (b) Spatial $R^2$ catchment performance in Generation 6 UK model Lat, Lon + WP + CC + HM + AHM (left) and regional models (right). A higher $R^2$ value is indicated by blue color, and a lower $R^2$ by red.





The NW region stands out for its relatively higher intra-catchment variability performance compared to other regions. The NW regional model achieved a mean $R^2$ of 0.49 and a median $R^2$ of 0.45, outperforming the SW in terms of both metrics. Although the aggregated regional $R^2$ values for the NW model (e.g., 0.75 in Generation 6) are lower than those of the SW and UK models, the NW model demonstrates a better ability to learn localized flood dynamics. This suggests that the NW

regional model prioritizes intra-catchment variability more effectively than inter-catchment generalization, highlighting a trade-off between capturing broad trends and localized dynamics. This is important to consider when evaluating the success of models. However, the NW model may be exhibiting a phenomenon known as Simpson's paradox (). This theory outlines the phenomena of seeing better model performance at aggregated scales compared to local scales, dependent on data partitioning. This is an important aspect to consider with any data-driven methods. Depending on how you manipulate and partition the

data, variability in results may be seen.

Overall, the UK model's strong $R^2$ reflects its ability to generalize across inter-catchment variability, benefiting from pooled data across regions. This is an established understanding in large-sample hydrology with machine learning (Kratzert et al., 2024a, 2019; Slater et al., 2024b). However, when considering extreme event magnitude data, intra-catchment variability remains poorly captured, as evidenced by relatively low catchment-level $R^2$ values in many regions in this study. The variability

in $R^2$ values across individual catchments highlights the inherent challenges of applying regional models to predict localized dynamics. Historical data constraints, particularly the limited number of extreme flood events available for many catchments, reduce the robustness of intra-catchment predictions. This catchment level variability, and discrepancy between scales model samples for extreme flood prediction, highlights the inherent challenges of applying regional models to predict localized catchment dynamics, where individual events are influenced by unique hydrological and meteorological conditions. The variability

in $R^2$ values within and across regions emphasizes the challenges of using regional models for catchment scale predictions. While aggregated regional-scale metrics often reflect strong overall performance, these metrics can obscure significant heterogeneity at the catchment scale. The under performance in certain catchments suggests that large-sample pooled models, while beneficial for generalizability and robustness (Kratzert et al., 2019, 2024b; Slater et al., 2024b, a), may oversimplify localized dynamics. Addressing this issue requires further feature engineering and tailored modeling approaches that account

for catchment-specific characteristics.

### 3.1.3 Explainable AI - Feature Importance

This section moves on from performance metrics, and focuses on process understanding. The SHAP summary plot results are presented for the best performing Generation 6 models at the UK and regional scale (UK, NW, SW), which exhibit statistically significant improvements over the Generation 1 baseline and reasonable catchment level performance. These results, shown

in Figure 6, provide valuable insights into the relative importance of features driving flood magnitude predictions for these regions.

The results for the UK, NW, and SW models demonstrate that precipitation and aridity dominate feature importance, which is expected and consistent with findings across previous works using random forest models for flood process analysis (Slater et al., 2024b; Coxon et al., 2024; Stein et al., 2021). In the UK model, the most influential features include aridity, precipitation on



the day of the event, base-flow index, runoff ratio, cumulative antecedent precipitation (one and two days prior), and potential evapotranspiration. Among these, aridity consistently merges as the leading predictor, reflecting its role in representing water availability and soil moisture conditions across diverse UK catchments.

However, while SHAP values provide valuable insights into model-predicted feature importance, they reflect associations rather than causal relationships. In our results, aridity consistently emerged as a top predictor, but with a positive influence on
flood magnitude predictions (see Figure. 6). This contrasts with the widely accepted physical relationship between increasing aridity and decreasing flood magnitudes (Coxon et al., 2024). This apparent contradiction highlights the need for caution when interpreting SHAP values. The positive association in SHAP results may stem from complex, non-linear interactions between aridity and other predictors, such as precipitation or catchment characteristics. Moreover, this may reflect event-specific patterns where arid catchments exhibit disproportionately large flood magnitudes under certain extreme conditions. This could further
be explained by the nature of the SHAP values, which consider interactions between features, such as precipitation. Moreover, SHAP analysis highlights non-linear relationships, and could be expressing the complex relationships between arid conditions limiting soil saturation, leading to faster runoff during intense precipitation events. While SHAP is a powerful explainable AI tool, its results must be interpreted alongside physical understanding, and caution must be taken when drawing conclusions. Depending on the interpretation tool used, different conclusions may be drawn (Slater et al., 2024a). This is critical to consider
and reflect upon when using machine learning for process analysis and relative importance ranking.

In the NW and SW regional models, the feature importance rankings largely align with the UK model but also reveal interesting regional nuances. Aridity remains the most important feature, followed by runoff ratio and precipitation on the day of the event. In the SW model, cumulative antecedent precipitation one day prior emerges as influential, reflecting the importance of previous days saturation in this region for extreme event magnitude generation. Variables such as maximum
elevation and mean temperature rank higher in importance in the SW model than in the UK model. This may suggest that the SW region's topographical diversity and regional climate influence,runoff generation and flood dynamics, highlighting the added value of building a region specific model to explore localized drivers of extremes. This result could be explained by higher elevations in SW catchments may exacerbate flood magnitudes by concentrating runoff, while temperature influences evapotranspiration and antecedent soil moisture.




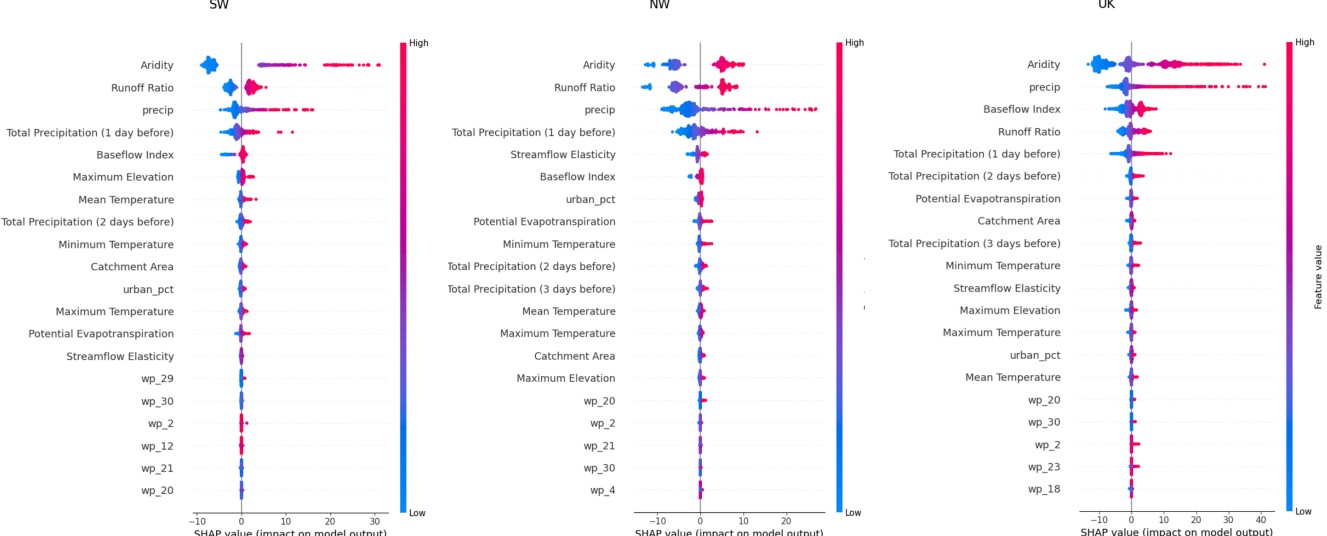

**Figure 6.** SHAP summary plot results for UK, NW, and SW Generation 6 models, illustrating feature contributions to each prediction. The y-axis lists the features ranked by their average absolute SHAP values, with the most impactful features at the top. The x-axis represents the magnitude and direction of the SHAP values, where positive SHAP values indicate an increase in the prediction and negative SHAP values indicate a decrease. Each point corresponds to a single prediction, with its color indicating the feature value (e.g., red for high predictor values and blue for low values).

The SHAP summary plot further supports the limited contribution of the WPs, which rank among the least important features across all models. These results reinforce earlier findings that including WP in this modeling framework, introduces redundancy or noise, reducing predictive accuracy.

Figure 7 displays the mean absolute SHAP values for the top five predictors for each region. For the best-performing models (UK, NW, SW), aridity, precipitation on the day of the event, and runoff ratio consistently emerge as the most important

predictors. This confirms the dominant role of hydrological and meteorological variables in flood magnitude predictions. In contrast, lower-performing models, such as those for ES and CEE regions, exhibit different feature importance rankings. For example, in the CEE model, base-flow index and maximum elevation are the top contributors, suggesting that these regions have distinct physical mechanisms driving extreme flood magnitudes. This may highlights how region specific drivers are also influencing predictive performance and suggests that models for ES and CEE regions may require additional feature

engineering to better capture local dynamics related to the highlighted important processes.

Interestingly, precipitation on the day of the event consistently ranks higher than antecedent precipitation (up to three days prior) in all regions, highlighting its direct impact on flood magnitudes in natural catchments. This finding suggests that, while antecedent conditions are important, the intensity and distribution of event-day precipitation may be very important for predicting extreme floods. These findings underscore the value of region specific models in uncovering localized drivers of



extreme flood magnitudes. This type of process analysis can be inform future model feature engineering to enhance predictive power.

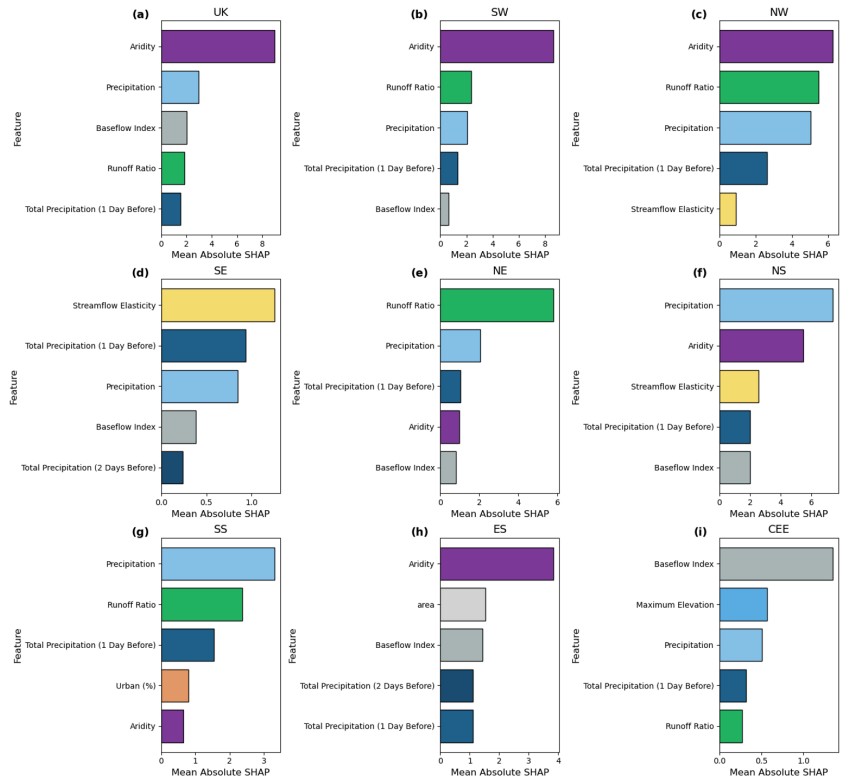

**Figure 7.** SHAP mean absolute values for the top 5 predictors for each of the UK and regional models. Further information on the predictors is in Table. 1.

## 3.2  Limitations and Future Recommendations

This study has several limitations that future research can address. One issue lies in the uncertainty associated with the extreme flow measurements, which are often associated with large observational errors due to rating curve extrapolation and

hydrometric limitations. Additionally, while the natural benchmark catchments were selected to isolate natural processes, their homogeneity may limit the variability in static catchment characteristics, therefore reducing these features as explanatory predictors. While these static descriptors such as catchment area, slope, and land cover are important for hydrological modeling, they do not capture dynamic processes such as soil moisture variation, saturation effects, or event-specific hydrometeorological conditions. Therefore, incorporating additional dynamic variables, such as groundwater, snowmelt, and soil moisture variables,

could significantly improve predictive performance. However, they are not currently available with the NRFA or CAMELS-GB time series. Another key limitation in this study is the uncertainty with precipitation data, it will not be completely perfect, and in a real-world forecasting application, there is not perfect knowledge of precipitation inputs.





The inclusion of WPs as predictors was motivated by their potential to provide predictable atmospheric information for flood magnitude generation, but their predictive value proved limited in this study. This could be due to their coarse spatial and temporal resolution, in comparison to localized extreme event data, leading to redundancy and noise in the model. Due to this scale mismatch between the WPs and flood magnitudes, future research should explore more refined representations of atmospheric dynamics, incorporating higher-resolution weather classifications or physically based indices that better capture synoptic-scale drivers of extreme floods. Moreover, further work into the optimum feature engineered temporal lag of WPs data to capture flood generation processes better and improve the explanatory power of WP data is required. Moreover, understanding the dynamic interplay between WPs, catchment characteristics, and types of antecedent saturation conditions would be a beneficial avenue for future research. Improved modeling of these interactions and key combinations of WPs and antecedent conditions, will enhance the ability to predict how different hydrometeorological conditions influence flood response. In particular, the projected increase in the frequency of high-impact weather patterns, such as WP 23 under climate change (e.g., Pope et al., 2021), underscores the urgency of refining predictive frameworks to account for evolving atmospheric and hydrological dynamics.

Model biases and regional variability also present challenges. Systematic underestimation in regions such as the SW and SE, contrasted with overestimation in NS, suggests that model performance is influenced by spatially heterogeneous hydrological processes and points to the need for tailored feature engineering and location-specific process analysis. Incorporating snowmelt dynamics in northern regions and improving the representation of antecedent precipitation dynamics in southern regions would be beneficial. Moreover, although national and regional scale performance metrics are useful, they are often masking substantial variability in performance at the catchment level. The poorer performance and high variability observed at the catchment scale is also likely due to the limited data size, an inherent issue with extreme event analysis. Future work should focus on multi-scale evaluation frameworks that can better capture local performance, and work on finding the optimum dynamic features with the best explanatory power for catchment scale processes.

Furthermore, while SHAP analysis provides valuable insights into feature importance, it is not without constraints. SHAP values can be sensitive to feature interactions. This can have implications on the consistency of interpretability across different studies. There should be cautious interpretation when drawing meaning from such methods. Overall, this study should be interpreted in the context of how the data was partitioned, selecting winter events only from near-natural catchments for extreme event magnitudes over the 99th percentile, resulting in limited observational coverage.

Future research should focus on several key areas to enhance the predictive accuracy and physical interpretability of flood forecasting models. One crucial area is region specific feature engineering, where locally tailored predictors are developed to better capture catchment characteristics across the UK. A more comprehensive representation of extreme flood generating mechanisms can be achieved by incorporating a wider range of catchment types, extreme flood events, and additional hydrometeorological variables, such as soil moisture, high-resolution precipitation data, and snowmelt estimates. This will help ensure that models are trained on diverse scenarios, leading to more reliable predictions across different hydrological regimes.





## 4    Conclusions

This study presents a comprehensive framework for applying ML models to quantify the contributions of different predictor sets in flood magnitude estimation across natural UK catchments, using synoptic-scale Met Office WPs as a predictor set. The UK-wide model demonstrated the highest predictive performance, achieving an $R^2$ of 0.84 in Model Generation 6, likely due to the benefits of pooling diverse regional data and learning inter-catchment variability. When $R^2$ was assessed at the catchment level within the UK-wide model, the intra-catchment performance varied substantially. Regional models, such as the NW and SW, also performed well at the overall aggregated $R^2$, and substantial variability in performance was observed at the catchment level as well. This highlights a key trade-off in large-sample modeling for extreme event magnitudes. Although pooling data from multiple catchments enhances overall model predictability by leveraging broader hydrological variability, it can obscure intra-catchment differences. When evaluating model performance at the individual scale, results can be significantly lower than the overall model score, reflecting localized processes that are not being captured by generalized models. This complexity underscores the inherent relationship between variability and predictability. While diverse datasets improve overall model robustness, achieving high accuracy at the catchment level for extreme event magnitudes requires accounting for fine scale controls, warranting further exploration of the processes driving extremes. This area of research is inherently limited by the small sample of events in the observational period, which is acknowledged as a key limitation in this study.

In terms of SHAP process analysis, hydrometeorological variables, including aridity, event-day precipitation, antecedent precipitation (3-days prior sum), and base-flow index, emerged as the most influential predictors across all models. These findings highlight the critical role of dynamic and localized features in capturing the complex interactions between atmospheric conditions and catchment-specific processes. In contrast, WPs contributed minimally as direct predictors, suggesting they do not sufficiently capture explanatory atmospheric information to enhance flood predictions. Although WPs have shown theoretical potential for linking large-scale circulation patterns to flood-generating mechanisms, spatial and temporal resolution mismatch with local extreme events may be restricting their effectiveness as standalone predictors. The regional differences in predictor importance, particularly in the SW and NW models, emphasize the need for regionally tailored feature sets to enhance model performance. For example, in the SW model, variables such as elevation and temperature played a more significant role, indicating that flood responses in this region may be strongly influenced by topography and climatic factors. These findings suggest that refining feature selection to account for catchment-specific hydrological and meteorological controls could further improve predictive accuracy.

Future developments in flood prediction should aim to integrate the advantages of large-sample modeling, which benefits from diverse datasets and increased generalizability, with regionally adapted approaches that capture the unique localized flood generating processes. In doing this, machine learning models can achieve both broader applicability and enhanced predictive skill across national, regional, and catchment scales.



*Code availability.* Code is available via GitHub repository https://github.com/emma-michelle/Can-weather-patterns-contribute-to-Predicting-Winter-Extreme-Flood-Magnitudes-Using-Machine-Learning.git)

*Data availability.* The datasets used in this paper are all publicly available. They can be downloaded from the National River Flow Archive,
the online Met Office HadUKP Regional Dataset and corresponding shapefiles, the CAMELS-GB dataset by Coxon et al. (2020), and the Met Office Weather Patterns by Neal et al. (2016)

.

## Appendix A: Supplementary Information 1

**Table A1.** Descriptions of the MO-30 weather pattern categories, reproduced from the datset provided by Neal et al. (2016)

| No. | Category | No. | Category |
|---|---|---|---|
| 1 | Unbiased northwesterly | 16 | Anticyclonic south-southeasterly with a high east of Denmark |
| 2 | Cyclonic southwesterly with a returning polar maritime airmass | 17 | Anticyclonic east-southeasterly with a high over Denmark |
| 3 | Anticyclonic southwesterly with a high pressure ridge over northern France | 18 | Anticyclonic southwesterly with a high over northern France |
| 4 | Unbiased westerly | 19 | Unbiased northerly with a low east of Denmark |
| 5 | Unbiased southerly with high pressure centred over Scandinavia | 20 | Cyclonic westerly with an intense low near Iceland |
| 6 | Anticyclonic Azores high extension towards the UK | 21 | Cyclonic southwesterly with a deep low south of Iceland |
| 7 | Cyclonic southwesterly with a low centred wester-northwest of Ireland | 22 | Cyclonic southerly with a low west of Ireland |
| 8 | Cyclonic westerly with a low centred near Shetland | 23 | Unbiased westerly and very windy in the north |
| 9 | Anticyclonic north-northeasterly with a high centred near Iceland | 24 | Cyclonic northerly with a low in the North Sea |
| 10 | Anticyclonic west-southwesterly with a slight Azores high ridge | 25 | Anticyclonic northerly with a high centred in the Irish Sea |
| 11 | Cyclonic with a low centred over southern UK | 26 | Cyclonic northwesterly with a low near Norway - very windy |
| 12 | Anticyclonic southerly with a high over Poland | 27 | Anticyclonic easterly with a high in Norwegian Sea |
| 13 | Anticyclonic northwesterly with a high southwest of Ireland | 28 | Cyclonic southeasterly with a low southwest of the UK |
| 14 | Cyclonic north-northwesterky with a low near southern Sweden | 29 | Cyclonic south-southwesterly with a deep low west of Ireland |
| 15 | Unbiased southwesterly, very windy in northwest Britain | 30 | Cyclonic west-southwesterly with a deep low southeast of Iceland |

*Author contributions.* EF designed the experiments, wrote the code, and conducted the analysis under the supervision of HC, MB, and LS.
MB, HC, and LS revised and edited the manuscript.

*Competing interests.* LS and MB are members of the editorial board of Hydrology and Earth System Sciences. The authors also have no other competing interests to declare.



*Disclaimer.* Copernicus Publications remains neutral with regard to jurisdictional claims in published maps and institutional affiliations

*Acknowledgements.* EF would like to thank the UKRI Natural Environmental Research Council, for funding the work conducted in this
paper. The award number is NE/S007474/1NE/S007474/1.

HC was funded by Natural Environment Research Council grant number NE/P018238/1, through a Leverhulme Trust Research Leadership
Award, and through the EERIE project (Grant Agreement No 101081383) funded by the European Union. University of Oxford's contribution
to EERIE is funded by UK Research and Innovation (UKRI) under the UK government's Horizon Europe funding guarantee (grant number
10049639). LS is supported by UKRI (MR/V022008/1). Views and opinions expressed are however those of the author(s) only and do
not necessarily reflect those of the European Union or the European Climate Infrastructure and Environment Executive Agency (CINEA).
Neither the European Union nor the granting authority can be held responsible for them.



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
