# Peer review of "Can Weather Patterns Contribute to Predicting Winter Flood Magnitudes Using Machine Learning?"

_EGUsphere, 2025_

## Author Comment (AC1)

**Title: Can Weather Patterns Contribute to the Prediction**

**of Winter Floods using Machine Learning?**

**Response to Reviewer 1**

We sincerely thank both reviewers for their thorough and constructive feedback. Their detailed comments have significantly helped us to plan amendments to the manuscript to improve the clarity, framing, and methodological rigor. We appreciate the recognition of the relevance of the topics addressed, as well as the thoughtful critiques regarding the use of weather patterns (WPs), the structure of our modelling framework, and the interpretation of results. In response, we are making substantial revisions to the manuscript. These include a reframing of the paper to better reflect its original contributions, particularly regarding the limited marginal value of WPs in flood magnitude prediction. We will improve the explanation of the methodological design and evaluation strategy and provide a more transparent and fair comparison between regional and national models. We have also clarified definitions, ensured consistent use of terminology, and addressed technical concerns raised in both the major and minor comments. Below, we provide responses to each comment. Throughout this document, our responses will be in blue text, and the reviewer comments in plain text.

R1C1: In the manuscript, if weather patterns contribute to predicting winter flood magnitudes was discussed using machine learning. To my knowledge, flood is mainly caused by intensive rainfall and antecedent soil conditions. In this study, they also considered these two factors and add some other variables. Some main questions are below.

Response: We thank the reviewer for highlighting this fundamental point regarding the drivers of flooding. We fully agree that flood generation is primarily governed by rainfall intensity and antecedent catchment conditions, such as soil moisture. These variables are central components of our study and are included in the feature set used for model training. Our specific focus on weather patterns (WPs) stems from the operational and research relevance of the 30-pattern classification developed by the UK Met Office (Neal et al., 2016). These WPs have been used in prior studies examining their links to precipitation extremes and droughts (e.g., Richardson et al., 2018; 2019; 2020) and are integrated into tools like the Met Office's Fluvial Decider for flood risk assessment. What sets our study apart is that we conduct the first large-scale, data-driven evaluation of the direct relationship between these WPs and observed fluvial flood magnitudes, using NRFA streamflow records. Our goal is to empirically assess whether large-scale atmospheric circulation patterns, often more predictable at longer lead time than local-scale variables, contribute meaningfully to flood prediction frameworks when more immediate forcings are also included. Ultimately, our results show that while WPs capture synoptic-scale atmospheric conditions, their marginal value in flood magnitude prediction is limited. This finding itself is important, as it quantifies the redundancy of large-scale indicators within data-driven hydrological models and helps refine which predictors offer true added value. We focus on weather patterns as the UK Met Office produced a set of 30 weather patterns, see Neal et al (2016). These weather patterns have been used in research associating them with precipitation and drought, and they are employed in a Met Office tool called Decider. We therefore conduct the first study relating these weather patterns directly to fluvial floods using the NRFA streamflow time-series data, to understand the relationship between them and flood magnitudes.

R1C2: Table 2, I cannot understand the relationship between total event count, number of catchments and catchment average event count.

Response: Thank you for this comment, we will improve the table to make this clearer to the reader. The total event count is the total number of fluvial flood magnitude events across all catchments. The number of catchments is the number of unique catchment IDs with events. The catchment average is the total events divided by the number of catchments.

R1C3: Line 154, 'pre-filtered to contain only extreme flood magnitude days', this will not ensure the flood event from beginning to the end.

Response: We appreciate the reviewer's comment and agree that filtering based solely on peak days does not capture the full duration or hydrograph shape of each flood event. However, our study is specifically designed to

focus on flood magnitude. By this, we mean the maximum daily flow or peak, associated with each extreme event, rather than the full flood hydrograph or event duration.

R1C4: Line 163, the categorize small, medium and large is not appropriate. Because in hydrology, there is a standard for definition of small, medium and large catchments.

Response: We thank the reviewer for this observation. While we acknowledge that hydrologists often use informal thresholds to classify catchment sizes, to the best of our knowledge there is no single, universally accepted standard for defining small, medium, and large catchments, specifically within the context of UK hydrology and the UKBN2 benchmark dataset. In this study, the terms "small," "medium," and "large" were used descriptively to indicate relative size classes, based on quantiles of the catchment area distribution within the UKBN2 dataset. This was done purely for interpretative convenience, rather than to imply strict hydrological classifications. We will clarify this point in the revised text.

R1C5: Line 278, the WP associated with the most extreme precipitation, does not necessarily translate to the WP associated with extreme flood magnitude days across UK regions.' I cannot understand the intrinsic relations between WP, extreme precipitation and extreme flood magnitude days.

Response: We thank the reviewer for raising this important point. The relationship between WPs, extreme precipitation, and extreme flood magnitudes is not direct or linear, and this is a central theme of our study. While previous research (e.g., Richardson et al., 2018) has shown that certain WPs are associated with heavy precipitation in the UK, extreme precipitation alone does not always result in extreme flooding. Flood magnitudes are influenced by several additional factors, including:

-   Antecedent catchment conditions
-   Catchment memory (e.g., the timing and accumulation of prior rainfall)
-   Catchment-specific properties (e.g., topography, land cover, drainage capacity)
-   Temporal structure of rainfall (e.g., short, intense bursts vs. prolonged moderate rainfall)

Even when a WP is associated with high rainfall in aggregate (through conditional probability), regional variability in hydrological response can lead to different WPs dominating the flood magnitude signal in different parts of the country. For example, a WP that causes widespread rainfall may trigger flooding in steep, saturated western upland catchments, but not in drier eastern regions with higher infiltration capacity. Topography plays a key role as well mountainous areas can respond more rapidly and intensely than flatter regions to the same synoptic forcing. Therefore, finding that a WP most associated with extreme precipitation is not necessarily the same WP most associated with peak flood magnitudes highlights the critical modulating role of other hydrological processes. This supports our conclusion that WPs, while useful for understanding large-scale atmospheric conditions, offer limited explanatory power for flood magnitudes when catchment-specific variables are not considered. We will revise the text around Line 278 to clarify this distinction more explicitly in the manuscript

R1C6: Line 363, CEE had the lowest baseline $R^2$ (0.28), and only the final $R^2$ of 0.37 in Generation 6 was statistically significant. Why the precision is so low? Are there any previous hydrological simulation in this region? Please compare this result with previous studies.

Response: We thank the reviewer for this important observation. The relatively low $R^2$ values for the Central and Eastern England (CEE) region reflect genuine modelling challenges in this area, which are likely driven by a combination of hydrological, data, and methodological factors. First, the CEE region is characterized by predominantly low-relief terrain, permeable soils, and mixed land use, leading to slower and more diffuse runoff responses to precipitation. These characteristics make peak flood magnitudes less directly tied to single rainfall events, and thus more difficult to capture using event-scale predictors alone. We are not aware of any published regional flood magnitude prediction models specifically for CEE. Higher uncertainty in this region could be explained by the baseflow-dominated hydrology and the difficulty in defining extreme event thresholds. Considering this, our $R^2$ value of 0.37, though modest, is consistent with regional hydrological characteristics and represents a statistically significant improvement over the baseline model. We will add further discussion in the manuscript to contextualize the CEE performance within existing literature and acknowledge the challenges of modelling in low-gradient, infiltration-dominated catchments.

R1C7: Line 370, 'The SE region's relatively lower sensitivity to antecedent precipitation and hydrometeorological inputs suggests that other factors, such as urbanization and engineered drainage systems, may dominate flood generation.' However, when you select the watersheds, they are not influenced by human activities.

Response: We appreciate this clarification and will revise the text to remove any implication that urbanization is a known factor in these specific sites.

R1C8: When using SHAP, you need to explain the definition of aridity, runoff ratio….

Response: We thank the reviewer for pointing this out. We agree that clear definitions of all predictor variables used in the SHAP analysis are essential for interpretability, especially for features like aridity index and runoff ratio that may vary slightly in definition depending on the context. In response, we will add a table to the manuscript (in the Methods or Supplementary Material) listing all variables included in the SHAP analysis, along with:

- Full variable names
- Units
- Definitions
- Data sources

For clarity, we define these variables as follows:

- Aridity Index: The ratio of potential evapotranspiration (PET) to precipitation (P) over the climatological period. A higher value indicates drier climatic conditions relative to available rainfall.
- Runoff Ratio: The ratio of mean annual streamflow (Q) to mean annual precipitation (P) at the catchment scale. This reflects how efficiently rainfall is converted to runoff, influenced by soil type, land cover, and infiltration capacity.

We will ensure these, and all other input variables are clearly described in the manuscript to improve reproducibility and interpretation.

R1C9: Line 490, 'The SHAP summary plot further supports the limited contribution of the WPs'. In traditional flood analysis, rainfall and soil moisture are the main contributors. We never consider WPs. In this study, WPs are focused, but still limited contribution. What is the innovation of this study?

Response: We thank the reviewer for this important question. Yes, traditional flood analysis rightly emphasizes precipitation and antecedent soil moisture as the dominant drivers of flood events. Our study does not challenge this understanding, instead, it aims to quantify whether atmospheric circulation patterns (in our case, synoptic scale WPs) provide any additional explanatory or predictive value when these direct drivers are already included in the model. The innovation of our study lies in the following key contributions:

1. First empirical test of WPs for flood magnitudes at national scale: While WPs have been linked to rainfall and drought in past studies (e.g., Neal et al., 2016; Richardson et al., 2018), this is the first study to evaluate their direct relationship with flood magnitudes across a large sample of UK catchments using machine learning and observational streamflow data.
2. Quantifying redundancy of large-scale indicators: By showing that WPs have limited marginal importance once local meteorological and catchment-scale features are included, we provide empirical evidence that these large-scale predictors are largely redundant in data-driven flood models. This is a useful insight for future feature selection and model design.
3. Challenging assumptions in operational tools: WPs are already being used in tools such as the Met Office's Decider. Our findings critically examine these assumptions and help define the limits of what WP-based reasoning can offer, especially for flood magnitudes rather than occurrence or likelihood.
4. Contribution to the growing interface between climate diagnostics and hydrology: The study advances understanding of how far ahead predictive signals can be leveraged, which is useful both for operational forecasting and long-term planning under climate variability.

In summary, while WPs were shown to contribute little additional predictive power in this specific context, our clear empirical evaluation of their role, and demonstration of their limited value, constitutes a meaningful and novel contribution to the field. We will revise the manuscript discussion to make this contribution more explicit.

R1C10: Line 501, 'Interestingly, precipitation on the day of the event consistently ranks higher than antecedent precipitation', actually, this is a common sense.

Response: We thank the reviewer for this observation. We agree that the dominant role of event-day precipitation in flood generation is well known in hydrology and should not be described as "interesting" in a way that implies novelty. We will revise the manuscript text at Line 501 to remove the word "interestingly" and instead frame the observation as a validation of model realism, consistent with standard hydrological expectations.

Thank you for your review!

---

## Author Comment (AC2)

**Title: Can Weather Patterns Contribute to the Prediction**

**of Winter Floods using Machine Learning?**

**Response to Reviewer 2**

We sincerely thank both reviewers for their thorough and constructive feedback. Their detailed comments have significantly helped us to plan amendments to the manuscript to improve the clarity, framing, and methodological rigor. We appreciate the recognition of the relevance of the topics addressed, as well as the thoughtful critiques regarding the use of weather patterns (WPs), the structure of our modelling framework, and the interpretation of results. In response, we are making substantial revisions to the manuscript. These include a reframing of the paper to better reflect its original contributions, particularly regarding the limited marginal value of WPs in flood magnitude prediction. We will improve the explanation of the methodological design and evaluation strategy and provide a more transparent and fair comparison between regional and national models. We have also clarified definitions, ensured consistent use of terminology, and addressed technical concerns raised in both the major and minor comments. Below, we provide responses to each comment. Throughout this document, our responses will be in blue text, and the reviewer comments in plain text.

Reviewer 2 Comments:

R2C1: The paper analyses two main relevant topics in Hydrology. The first one is the predictability of extreme flooding events. The second topic is the difference between national (UK) and regional models. The main finding from the first one is that WP is not relevant for prediction, mainly because other attributes and forcing already share the same information. In the second topic, the national model exhibits the overall best performance, but with considerable variability between some regional models, indicating that, in many cases, regional models capture the dynamics of the region more effectively. The results align with other research; however, some concerns arise from the framework and the presentation of the results.

Main comments

The title is not aligned with the framework and the results. The author used a progressive feature incorporation to explore the benefit of having them in the model. From this analysis, WP was not relevant or caused a deterioration of the performance in most of the regions. Therefore, the author should not use them; however, they insist on using them across the entire paper despite of the no-value. The same happens with other features in generations 2, 3, and 4. My suggestion is to reframe the title and the paper toward the characterization of the extreme events through different types of models, and keep just the relevant features, which will help to have a better interpretability of the results.

Response: We thank Reviewer 2 for the thoughtful and critical evaluation of our manuscript. We greatly appreciate the recognition of the importance of our central themes, such as the predictability of extreme flooding events and the comparative performance of national versus regional models. We acknowledge the concern that the framing of the paper did not fully align with the results, particularly regarding the role of weather patterns (WPs) and other features with limited or negative marginal value. Our revisions aim to address this misalignment and improve the interpretability of the work. Specifically, we will:

- Revise the title and abstract to emphasize that the study tests and ultimately questions the added value of large-scale climate indicators like WPs. The revised title will reflect the papers contribution better, regarding the model framework.
- Clarify throughout that WPs and other weak predictors were retained as part of a systematic evaluation of their marginal contribution relative to hydrometeorological and catchment-scale predictors.
- Ensure fairer comparisons between UK-wide and regional models by using matched catchment sets and consistent evaluation metrics.
- We will provide a version of the final model without the WPs to address this point further.

We believe these adjustments strengthen the manuscript by improving clarity, transparency, and alignment between framing and findings. Importantly, our decision to include WPs is not contradictory to the results

but reflects our aim to demonstrate and quantify the limited value of synoptic-scale indicators in this modelling context.

R2C2: Another concern is how the results are presented. In Figure 4, the UK model is presented as the best model (Generation 6). This would mean the model should outperform regional models at least in 50% of the cases. However, Figure 5a shows us that the UK model has a lower median than NW and NE. That is possible given the variability in the model, as the authors describe; however, the figure may be misleading because to have a real comparison, the authors should compare the UK model with each regional model by selecting the same catchments in both, which appears not to be the case. In fact, Figure 5b shows that the concentration of blue dots is higher in the UK model, which is consistent with Figure 4. The author describes the Simpson's paradox as the problem; however, they forgot that they are responsible for having a fair comparison and splitting of the data. Therefore, they should avoid the paradox, which, from the differences between Figures 4 and 5, is not the case.

I suggest a major revision of the paper, given that they need to reframe the paper to the actual results and check that all the results are presented in a fair way to avoid misleading.

Response: We thank the reviewer for raising this crucial point regarding fairness in the comparison between the national (UK) model and the regional models. We fully agree that clear and valid comparison requires consistent sample sets across models and metrics calculated over matched catchments and time periods. We agree that a valid comparison requires evaluating the models on identical sets of catchments and time periods. In the original manuscript, Figures 5a and 5b compared distributions and spatial patterns of performance. However, as the reviewer points out, these comparisons were based on different catchment pools (UK-wide vs. region-specific), which may lead to misleading impressions.

To address this, we will:

1. Recalculate all comparisons using matched catchments, e.g., for each region, evaluate the UK model only on the subset of catchments used in that region's model.
2. Replace Figures 5a and 5b with a new multi-panel figure showing, for each region, side-by-side comparisons of the regional model and the UK model restricted to the same catchments. These panels will include both:
   (a) Boxplots of regional vs. UK (subsampled) performance, and
   (b) Catchment-level $\Delta R^2$ scatterplots (UK and regional), to illustrate differences transparently.
3. Revise captions and text to clearly state that performance metrics are always computed on a per-catchment basis and that comparisons are made on matched samples. We will also remove the reference to Simpson's paradox, instead explaining the discrepancy between Figures 4 and 5 because of aggregation choices.

We believe these changes will ensure fairer and more transparent model comparisons and improve clarity in the presentation of results. We agree that this correction will improve the fairness and clarity of our model evaluation framework. These changes will be reflected in both the Methods and Results sections.

Minor comments:

R2C3: Line 24          Check reference (?)

Response: Thank you. We will amend this.

R2C4: Line 29-30          What about events with no high intensity but longer duration?

Response: We thank the reviewer for this observation. While our text referenced key simplified mechanisms of flood generation such as intense rainfall and saturated soils, we agree that long-duration, lower-intensity rainfall events also play a critical role, especially in large, lowland, or more permeable catchments where flood response integrates over longer accumulation periods. To address this, we will revise the text to acknowledge this mechanism more explicitly and clarify that while high intensity rainfall is often emphasized, flood generation may also result from prolonged rainfall events that gradually saturate the soil and exceed drainage capacity.

R2C5 and R2C6:

Line 43          Any idea why they have been used before?

Line 45           How have other studies been done before? You should present the baseline to have a clear benefit of your approach.

Response: In response to the comments regarding line 43 and 45 together: We thank the reviewer for this question. These Met Office 30 synoptic scale WPs have been used in past studies usually to characterize broadscale atmospheric conditions that influence precipitation and drought, or to classify regimes relevant to hydrological modelling. For example, the work by Richardson et al (2018, 2019, 2020). More recently, they have been incorporated into decision-support tools such as the Met Office's Decider which has applications such as Fluvial Decider (Richardson et al, 2020) which uses WPs to assess flood risk probabilistically, based on the WPs association with UK regional precipitation distributions. However, despite their use in such frameworks, WPs have rarely been integrated into data-driven flood magnitude models, in large-sample machine learning contexts. This is possibly due to the added complexity of multi-scale feature interactions, limited precedent for encoding WPs in ML models, and a focus on more direct hydrometeorological forcings such as the prediction of precipitation. We will revise the manuscript to clarify both the rationale for their past use and the gap that our study addresses by formally evaluating their added value in a predictive ML framework. Thank you.

R2C7: Line 87          What about the look at table approach implemented in RF? How can this go against your results?

Response: We thank the reviewer for raising this point. By "look-up table behaviour," we refer to the tendency of Random Forests to approximate predictions by memorizing specific combinations of feature values observed during training. The model can act like a stored table of rules, rather than learning generalizable relationships, especially when input patterns are highly repetitive or when training and testing distributions are very similar. However, in our case, we believe this effect is limited for two reasons:

1. Event rarity and spatial diversity: Our focus on flood events above the 99th percentile ensures that the input patterns are highly variable, with relatively few repeated combinations. This reduces the chance of the model relying purely on memorized lookup rules and instead requires it to learn generalizable associations between predictors and outcomes.
2. We employed temporal validation. The performance on the held out catchments in time indicates that the model is not simply memorizing event specific combinations but is capturing transferable patterns.

We do agree this is a valid concern and could be more robustly considered in the modelling approach and is critical to consider when interpreting the model's decision structure. To address this, we will clarify in the manuscript that while RF models can mimic look-up behaviour, our use of cross validation helps ensure the model learns meaningful, generalizable relationships. We will also mention this as a limitation and suggest exploring additional model architectures in future works.

R2C8: Line 98-99          Why do you need to cluster the model if the RF architecture already does clustering? Why do you think your clustering can do it better than RF?

Response: We thank the reviewer for this insightful question. While Random Forests inherently partition feature space through recursive splitting, this process is purely data-driven and does not explicitly capture hydrological or climatic regional structures. Our use of predefined regional models (a form of spatial classification) serves distinct purposes beyond RF's internal partitioning:

1. Improved interpretability: Grouping catchments into established hydroclimatic regions allows us to better interpret geographic patterns in predictor importance and model performance, which would not be possible if relying solely on RF's implicit splits.
2. Hydrological relevance: Regional models provide a framework to examine region-specific flood-generation processes and to test whether localized models better capture these processes than a pooled UK-wide model.
3. To illustrate if regional specific processes are better captured by local (regional) models or a UK wide pooled model.

We will revise the manuscript to clarify that regional grouping is a deliberate modelling design to support region-specific interpretation, and not a substitute for RF's internal partitioning.

R2C9: Line 151-152   Is this analysis linear? If this is the case, what are the implications of that analysis?

Response: We thank the reviewer for this question. To clarify, the analysis described in this paragraph is not linear and does not involve fitting any regression or parametric model. It is a non-parametric, frequency-based assessment of conditional probabilities, specifically:

P (WP | flood) refers to the probability that a given WP occurred, given that a flood magnitude event was identified. This approach is purely descriptive and intended to identify which weather patterns are most frequently associated with extreme flood events, both nationally and within regions. We also consider lagged WPs (up to three days prior) to account for potential lead time influence on flood magnitudes. We appreciate the implications of this analysis are interpretive rather than predictive, and it was our aim to highlight patterns of co-occurrence of synoptic patterns and floods, but we acknowledge this does not quantify causal or linear relationships. We will revise the manuscript text to explicitly state that this is not a linear or model-based analysis, to avoid any confusion.

R2C10: Line 169          Check reference (year?)

Response: Thank you. We will amend this.

R2C11: Table 3           Add all variables considered in each category and the abbreviation used per group.

Response: We thank the reviewer for this helpful suggestion. To improve clarity and reproducibility, we will expand Table 3 to include the full list of variables introduced at each model generation, abbreviations used in the input dataset or SHAP plots, and brief descriptions and units where appropriate. This additional detail will either be added directly to the manuscript or included as a supplementary table.

R2C12: Line 173-175   This is a result, so it should not be in the methodology.

Response: Thank you. We will amend this.

R2C13: Line 179          Why did you use a two-period splitting when in ML, three periods is the common practice to avoid overfitting and leaking information?

Response: We thank the reviewer for this valuable point. We used a two-period temporal split, training (1969–2010) and testing (2011–2021) to ensure that the model was validated on future, unseen data in a time-consistent manner. While a three-period split (train–validation–test) is standard in many ML workflows, our setup reflects the structure commonly used in hydrological ML research, particularly when extreme events are sparse. To prevent overfitting and leakage: (1) hyperparameter tuning was performed using RandomizedSearchCV with cross-validation entirely within the training period, and (2) the test set was untouched during both training and tuning and was used only for final evaluation and SHAP interpretation. This approach respects the temporal integrity of the data and avoids forward-looking bias. However, we acknowledge that it does not fully isolate the hyperparameter search process from model evaluation. We will revise the manuscript to clarify this distinction.

R2C14: Line 183-184   This is not completely true because you used this period for the hyperparameter search, so it is not unseen data.

Response: We thank the reviewer for raising this concern and appreciate the opportunity to clarify. The test period (2011–2021) was not used for hyperparameter search. Hyperparameters were optimized exclusively within the training period (1969–2010) using RandomizedSearchCV and sensitivity analysis, after which a single parameter set was fixed and applied consistently across all model generations. Therefore, the test period remained entirely unseen during both training and hyperparameter tuning. Our use of temporal splitting ensured that evaluation reflected a realistic forecasting setting, where models are validated on a future period not available during model development. We will revise the manuscript to make this distinction clearer. Specifically, that while the test set was unseen in all respects, hyperparameter selection was conducted within the training set only, not across both periods.

R2C15: Line 197         The metric can take negative values.

Response: We acknowledge that the statement in the manuscript was incomplete. While $R^2$ is commonly described as ranging from 0 to 1 in well-performing models, it can in fact take on negative values if the model performs worse than simply predicting the mean of the observed values. We will revise the text accordingly to reflect this more precise and technically accurate interpretation.

R2C16: Figure 2         Numbers must be located at the center of the column (x-axis)

Response: Thank you. We will amend this.

R2C17: Figure 3         You should uniform the text sizes in the figure.

Response: Thank you. We will amend this.

R2C18: Line 293         Could it be that WP30 is associated with the duration or total volume of the event? From figure 1, it is clear that WP30 is related to how big the WP is and the overlapping between the low-pressure area and the UK boundary.

Response: We appreciate the reviewer's thoughtful observation and agree that WP30's spatial extent and persistence may be more closely related to event duration or total rainfall volume, rather than peak magnitude alone. As correctly noted, Figure 1 WP30 is a broad, cyclonic system and this type of system is often associated with widespread and prolonged precipitation across the UK. This may indeed explain why WP30 is frequently associated with flood days yet not consistently linked to the highest peak magnitudes (Figure 3a). These findings suggest that WP30 could drive lower intensity but longer duration events, especially in larger catchments where prolonged rainfall is a more important factor for flood generation. While this study focused on flood magnitude days in near-natural UK catchments, we acknowledge that evaluating event duration or cumulative rainfall volume would offer additional insights. We will include this interpretation in the revised discussion and suggest it as an avenue for future work. To reflect this, we will add a clarification in the manuscript regarding the frequent association of WP30 with flood days, despite not yielding the highest magnitudes, may indicate that it plays a greater role in driving long-duration or high-volume events, particularly in larger catchments. Future research should explore this hypothesis using event-based cumulative rainfall or flood volume as alternative metrics.

R2C19: Line 324         Check reference (?)

Response: Thank you. We will amend this.

R2C20: Line 381         Latitude and longitude do not have a hydrological meaning other than specific coordinates in space. For that reason, they are mainly used as an address to identify each catchment, and are much more specific than any catchment attributes. If you want to have more hydrological meaning, latitude and longitude should not be used as attributes.

Response: We thank the reviewer for highlighting this important conceptual distinction. We agree that latitude and longitude are not inherently hydrological variables, but rather proxies for location. We will ensure to explain this in the revised text. However, in this study, they were intentionally included in Generation 1 to serve as a baseline representation of spatial variability, capturing geographically dependent influences that may relate to underlying hydrological processes. Their strong performance in early generations likely reflects this implicit spatial encoding, since they are unique per catchment and can indirectly capture broader scale regional patterns. We acknowledge that interpreting latitude and longitude as meaningful hydrological predictors is limited, and we will clarify in the manuscript that these features primarily serve as spatial identifiers. We will make an amendment in the manuscript to explain this, that the inclusion of catchment characteristics (Generation 4) generally resulted in $R^2$ values worse than or comparable to the baseline model (Generation 1), suggesting that latitude and longitude though not hydrological in nature, act as strong spatial identifiers. They may implicitly capture regional variability or confounding spatial gradients, thereby limiting the marginal value of static catchment attributes when added separately. We also plan to make a note in the Limitations and Future Work section, mentioning that future studies should aim to disentangle spatial encoding from physical process representation.

R2C21: Line 412          Please describe the intra/inter concept before using it.

Response: Thank you for this helpful suggestion. We agree that the distinction between intra-catchment and inter-catchment variability should be clearly defined beforehand. We will revise the text prior to Line 412 to introduce and define these terms explicitly. The following clarification will be added in this context to describe intra-catchment variability as referring to the variation in flood magnitudes within a single catchment over time (how well the model captures temporal changes within one location), whereas inter-catchment variability refers to the differences in flood responses between different catchments.

R2C22: Line 423-424   I agree that specific features could help, however, given the use of latitude and longitude, it will be hard the find features that are more specific than those. This is one of the problems of using attributes that are not hydrologically meaningful.

Response: We appreciate this insightful comment. We fully acknowledge the limitations of using latitude and longitude as input features, especially as they serve primarily as spatial locators rather than hydrologically meaningful predictors. However, in our baseline model (Generation 1), they served as a practical proxy for inter-catchment variability, which is otherwise captured through more complex, static catchment descriptors. We will also explain in the revised manuscript that latitude and longitude are essential for the model to help interpret the WPs, as WPs will influence different spatial locations in different ways. Subsequent generations of the model were designed to incrementally introduce more physically meaningful attributes (to move toward a more interpretable and hydrologically grounded framework). We also will add a sentence to the discussion noting that the strong predictive performance of latitude and longitude may suggest that better spatially distributed catchment attributes are needed, and that future work should prioritize physically interpretable spatial features that reduce the need for spatial placeholders like coordinates.

R2C23: Line 429          This goes against the findings you already mentioned and figures 4 and 5b.

Response: We thank the reviewer for this observation. We agree that the original statement appeared contradictory when compared with Figures 4 and 5. As noted above, we will remake Figure 5a (and 5b) to ensure that the UK and regional models are compared on the exact same set of catchments, thereby providing a fairer basis for comparison. We will then revise the corresponding text at Line 429 to ensure it is fully consistent with the updated results. This will clarify the nuanced relationship between the UK and regional models without overstating either side.

R2C24: Figure 5b          Could you replace the regional figure with the difference between the UK and the regional one?

Response: Thank you for this helpful suggestion. We agree that showing the difference between the UK and regional model $R^2$ performance at the catchment level would better highlight areas where each approach outperforms the other. In response, we will update Figure 5b to display the spatial difference in $R^2$ values between the UK model and the corresponding regional model for each catchment.

R2C25: Line 437          Check reference. I agree that it is an important issue; however, you are responsible for the framework to avoid that. How are you calculating the median for the local and global metrics? You should always be calculating the metric per catchment and then computing the median independent of the model, and all over the same period and the same group of catchments.

Response: We agree that the reference to Simpson's paradox was incomplete and insufficiently justified in the original manuscript. We have now removed the reference to Simpson's paradox entirely, as its use was not clearly supported by the metric aggregation framework we implemented. Instead, we have rephrased the sentence to focus on the discrepancy between aggregated (regional) and disaggregated (catchment-level) model performance, which better reflects the issue at hand. We also appreciate the request for clarification on how median and mean metrics were computed. In response, we have explicitly stated in the revised manuscript that $R^2$ values were computed at the individual catchment level first, based on the model's test set predictions. Aggregated statistics (mean and median) for regional and UK models were then calculated from this catchment-level distribution, ensuring consistent comparison across models, regions, and time (2011–2021).

R2C26: Line 447-449   Could this issue be just part of the overfitting, given the low number of events? I think more work must be done to clarify how the R2 is calculated. Maybe this conclusion is just an artifact of the computing method.

Response: Thank you for this important comment. We acknowledge that the limited number of extreme flood events per catchment presents a challenge and may indeed contribute to overfitting or unstable $R^2$ estimates, especially in smaller or more sparsely observed catchments. To address this, we have clarified in the manuscript that $R^2$ is computed at the catchment level only for those with more than 10 test set events, as noted in Section 2.5. This threshold was chosen to improve the reliability of performance metrics while acknowledging the constraints of event-based data. Additionally, all aggregated $R^2$ statistics (mean and median) presented in Figures 4 and 5 are based on these individual catchment-level results, ensuring consistency across all models and regions. We agree that some of the observed variability in performance may reflect statistical artifacts arising from data scarcity or overfitting, rather than meaningful differences in model generalization. To reflect this uncertainty, we will explain that given the relatively low number of extreme events in many catchments, some of the observed intra-catchment performance variability may stem from statistical artifacts or overfitting. These results should be interpreted cautiously, and future work should explore alternative validation schemes.

R2C27: Line 465          Given the use of latitude and longitude in all the generations, and the high performance with the first generation, it is weird that those features do not appear as one of the most important features in SHAP.

Response: We appreciate the reviewer's observation. The absence of latitude and longitude in the SHAP summary plots is due to a deliberate decision to omit them from the visualizations to highlight physically interpretable predictors with hydrological or meteorological meaning. We acknowledge that latitude and longitude played a central role in Generation 1 and encode important spatial variability and unobserved catchment characteristics. However, because they are spatial identifiers rather than process-based features, including them in the SHAP plots could risk misinterpretation of their importance as physical drivers of flood magnitude. To avoid confusion, we will explicitly clarify this decision in the manuscript text (Section 3.1.3) and in the figure captions. We are also happy to provide supplementary SHAP plots including latitude and longitude for transparency.

R2C28: Line 486-489   Many of these variables have other collinear variables, so you should try to prune the model with more independent variables. This way, you will have stronger relationships with the attributes.

Response: We appreciate this valuable suggestion. We agree that multicollinearity among input features introduces redundancy and may weaken the interpretability of feature importance scores. In our current framework, we did not explicitly perform feature selection or collinearity pruning, as the Random Forest (RF) algorithm is often relatively robust to multicollinearity due to its ensemble structure and random feature sampling. For example, as discussed in Linder et al., 2022. However, we recognize that removing or grouping highly collinear features could improve both interpretability and reduce the noise in SHAP-derived importance rankings. In response to the reviewer's comment, we will now include clarification in Section 3.1.3 of the manuscript. We will explain while Random Forest models are relatively tolerant to multicollinearity, future work could consider pre-modelling strategies to assess feature redundancy. We will also go one step further, and we can perform a collinearity test, and quantify the effect this has on the results. By adding this plot to the revised manuscript, we aim to address this comment further.

R2C29: Line 501          Precipitation is well known in hydrology as one of the most important variables, so I would not say that this is interesting.

Response: We thank the reviewer for this observation. We agree that the dominant role of event-day precipitation in flood generation is well known in hydrology and should not be described as "interesting" in a way that implies novelty. We will revise the manuscript text at Line 501 to remove the word "interestingly" and instead frame the observation as a validation of model realism, consistent with standard hydrological expectations.

R2C30: Line 505          That the UK model has different importances does not mean it does not capture the local variability of the importance. The UK's importance is just overall more important. You can think of the

regional models as specific branches of a big tree. In that case, each branch has different importance because they are independent. Therefore, this comment is unfair to the UK model.

Response: We thank the reviewer for this insightful comment. We agree that differences in SHAP importance between the UK and regional models should not be interpreted as the UK model failing to capture local variability. Rather, both UK and regional models report global feature importance, but computed over different sample sets: the UK model reflects importance aggregated across all catchments, while the regional models reflect importance aggregated within their respective subsets. We will revise the manuscript to make this distinction clearer. Specifically, we will emphasize that regional models highlight drivers that are dominant within specific hydroclimatic contexts, whereas the UK model emphasizes features that are influential across the full range of catchments. In this sense, differences between UK and regional SHAP values are best viewed as scale-dependent expressions of importance, not as a limitation of the UK model. We also appreciate the "branches of a tree" analogy suggested by the reviewer and will adapt this in the revised text to illustrate the nature of interpretability across spatial scales.

R2C31: Line 517          There are ways to quantify uncertainty. In fact, RF has already un uncertainty quantification that you did not use (ensemble). Therefore, more effort should be made to consider it.

Response: We agree with the reviewer that Random Forests offer a built-in mechanism for estimating predictive uncertainty, for example by analysing the distribution of predictions across trees in the ensemble. In this study, we focused on performance metrics ($R^2$, PBIAS) and interpretability via SHAP to understand the role of different feature sets in flood predictions. We acknowledge that incorporating uncertainty quantification, e.g., using the variance or quantiles of tree predictions could add important insight, particularly for extreme events where uncertainty is inherently high. We will provide the uncertainty information in the revised manuscript to address this comment. Thank you.

R2C32: Line 518-530   Why should researchers spend time refining WP if they had zero importance? Maybe it could be more beneficial to refine catchment attributes that you have already proved are important.

Response: We thank the reviewer for this important question. Our SHAP analysis indeed shows that the categorical WPs, as applied here, contributed little predictive value compared with hydrometeorological or catchment attributes. We fully agree that refining catchment descriptors remains a highly valuable direction for improving model performance. However, we believe there is rationale for exploring refined WP formulations in future work. The WP types used here are categorical and relatively coarse, which may dilute their signal. Alternative approaches could test higher temporal resolution circulation types, or tailoring WP definitions to UK-scale hydrological regimes. While the current formulation was not informative, refinements may prove useful in contexts like seasonal forecasting or climate change attribution. We will revise the manuscript to clarify this point and emphasize that refining both catchment attributes and atmospheric indicators are complementary avenues for advancing predictive skill.

R2C33: Line 543-544   However, you defined the framework, why did you not change the percentile to 95% or another value to have more data? Do you think that the important features or relationships would change if you use other percentiles?

Response: We appreciate the reviewer raising this point. Indeed, the choice of the 99th percentile threshold was a deliberate one, intended to focus the analysis on the most extreme and impactful flood magnitude events. However, we fully acknowledge that threshold selection can influence both sample size and the processes that appear most important in driving those extremes. To explore this, we conducted a preliminary sensitivity analysis using lower percentile thresholds (e.g., $90^{th}$ and $95^{th}$ percentiles). While the lower thresholds enabled a larger data sample size, we observed that the inclusion of WPs remained low in predictive importance across thresholds, and the key hydrometeorological drivers (e.g., aridity, precipitation, runoff ratio) remained dominant. However, as expected, subtle differences in feature rankings emerged, suggesting that process importance does vary with event severity. This would be a valuable direction for future work and is a project of its own. Therefore, we opted to present the 99th percentile winter results to maintain consistency with extreme event definitions used in related literature, and to emphasize the most societally relevant floods. We agree this topic warrants deeper exploration and have highlighted it in our revised discussion as a potential avenue for future research. We will also include some summary plots showing the outcomes of this sensitivity analysis to threshold in the supplementary material, to address this comment further.